

# Reevaluating the black carbon in the Himalayan and Tibetan Plateau: concentration and deposition

Chaoliu Li[1,3,4*], Fangping Yan[3], Shichang Kang[2,4], Pengfei Chen[2], Xiaowen Han[1,5],

Zhaofu Hu[2,5], Guoshuai Zhang[1], Ye Hong[6], Shaopeng Gao[1], Bin Qu[3], Mika

Sillanpää[3,7]

[1]Key Laboratory of Tibetan Environment Changes and Land Surface Processes, Institute of Tibetan

Plateau Research, Chinese Academy of Sciences, Beijing 100101, China;

[2]State Key Laboratory of Cryospheric Sciences, Northwest Institute of Eco-Environment and

Resources, Chinese Academy of Sciences, Lanzhou, 730000, China;

[3]Laboratory of Green Chemistry, Lappeenranta University of Technology, Sammonkatu 12,

Mikkeli 50130, Finland

[4]CAS Center for Excellence in Tibetan Plateau Earth Sciences, Beijing 100101, China;

[5]University of Chinese Academy of Sciences, Beijing 100049, China

[6]Institute of Atmospheric Environment, China Meteorological Administration, Shenyang 110166,

China

[7]Department of Civil and Environmental Engineering, Florida International University, Miami,

FL-33174, USA

*Correspondence to:* C. Li (lichaoliu@itpcas.ac.cn)

**Abstract:** Black carbon (BC) is the second most important warming component in the

atmosphere after $CO_2$. The BC in the Himalayan and Tibetan Plateau (HTP) has shaped

the evolution of the Indian Monsoon and accelerated the retreat of glaciers, thereby

resulting in serious consequences for billions of Asian residents. Although a number of

related studies of this region have been conducted, the BC concentration and deposition

indexes remain poorly understood. Because of the presence of arid environments and

the potential influence of carbonates from mineral dust (MD), the reported

concentrations of BC from the HTP are overestimated. In addition, large discrepancies

in the deposition of BC have been reported from lake cores, ice cores, snowpits and

models. Therefore, the actual BC concentration and deposition values in this sensitive

region must be determined. A comparison between the BC values of acid (HCl)



fumigated and original aerosol samples from the HTP showed that the BC concentrations previously reported for the Namco station (central part of the HTP) and the Everest station (northern slope of the central Himalayas) were overestimated by approximately 47±37% and 35±26%, respectively, because of the influence of

carbonates from MD. Additionally, the organic carbon (OC) levels were overestimated by roughly $22\pm10\%$ and $22\pm12\%$ for the same reason. Based on previously reported values from these two areas, we propose that the actual BC concentrations at the Namco and Everest stations are 44 ng m$^{-3}$ and 164 ng m$^{-3}$, respectively. Second, a comprehensive comparison of the BC deposition levels obtained via different methods

indicated that the BC deposits derived from lake cores of the HTP were mainly caused by river sediments transported from the lake basin as a result of climate change (e.g., increases in temperature and precipitation), and fewer BC deposits were related to atmospheric deposition. Therefore, previously reported BC deposition levels from lake cores overestimated the atmospheric deposition of BC in the HTP. Correspondingly, BC

deposition derived from snowpit, ice core and model from the HTP were not only agree very well with each other, but also were close to those of other remote areas (e.g., Arctic), implying that the BC deposits calculated from these three methods reflect the actual values. Therefore, based on reported values of snowpits and ice cores, we propose that the BC deposits of the HTP range from 10 mg m$^{-2}$ a$^{-1}$ to 25 mg m$^{-2}$ a$^{-1}$,

with high and low values appearing along the fringes and central areas of the HTP, respectively. The adjusted BC concentration and deposition values in the HTP observed here are critical for performing accurate evaluations of other indexes of BC such as atmospheric distribution, radiative forcing and chemical transport in the HTP.

**Key words:** the Himalayan and the Tibetan Plateau; black carbon; concentration;

deposition; glacier region

## 1. Introduction

The Himalayan and Tibetan Plateau (HTP) is the highest plateau in the world and represents the source region of approximately ten large rivers in Asia; moreover, this

region is sensitive to climate change (Huang et al., 2008;Kang et al., 2010;Bolch et al.,





2012). Black carbon (BC) deposits in and around the HTP have been found to play key roles in climate change patterns in the HTP and Asia and cause atmospheric warming (Xu et al., 2016;Ramanathan and Carmichael, 2008;Lau et al., 2010), promote HTP glacial retreat (Xu et al., 2009b;Qu et al., 2014), alter monsoon system evolution (Bollasina et al., 2008) and modify fresh water supplies to billions of residents across Asia. Therefore, BC in the HTP have become an important research topic in recent years. Lots of studies on BC concentrations, carbon isotopic compositions, seasonal variations, sources, transportation and radiative forcing have been conducted in recent decades (Zhao et al., 2013;Ming et al., 2010;Xu et al., 2009b;Xu et al., 2009a;Cong et al., 2015;Li et al., 2016b;Lu et al., 2012;Zhang et al., 2015;Marinoni et al., 2010;Rattigan et al., 2010;Kaspari et al., 2011). However, the above studies present limitations because of unique environments found in the HTP (e.g., high mineral dust (MD) concentrations in aerosols and catchment inputs to lake sediment). Therefore, the above studies should be reinvestigated to better define the actual BC values. In this article, we discussed the concentration and deposition of BC in the HTP, which represent basic input data that can support other important studies on the sources, radiative forcing patterns and chemical transport of BC in this region.

Concentration represents a basic parameter used in BC-related research, and the thermal-optical method is one of widely used methods for measuring BC concentrations in the aerosols of the HTP (Zhao et al., 2013;Ming et al., 2010;Cong et al., 2015;Li et al., 2016d). An important factor that influences the accurate measurement of BC concentrations via this method is the presence of carbonates (inorganic carbon (IC)) from MD, which is ubiquitous in the atmosphere. IC can also emit $CO_2$ during periods of temperature increase, thus causing the overestimation of total carbon (TC=organic carbon (OC) + BC) and BC concentrations (Karanasiou et al., 2010). Hence, IC is generally excluded in carbonaceous aerosol studies (Bond et al., 2013). However, few studies of the HTP have considered the contributions of IC to TC and BC because a research study conducted in the United States over ten years ago (Chow and Watson, 2002) concluded that IC can be neglected in studies of TC and BC for mid-latitude aerosols because of its far lower concentrations relative to the concentrations of TC and



BC.

This conclusion cannot be blindly applied to other areas because of the complexities mid-latitude environments around the world (e.g., arid areas and deserts with heavy dust storm events). For example, IC accounts for roughly 8% (Cao et al.,

2005) and 10% (Ho et al., 2011) of the TC particles with diameters less than 2.5 μm (PM$_{2.5}$) during dust storm periods in northern China. Similar phenomena have been found for both PM$_{2.5}$ and total suspended particle (TSP) samples in southern Europe (Sillanpää et al., 2005;Perrone et al., 2011). Because TSP contains more MD than PM$_{2.5}$, it should contain a higher ratio of IC. In that study mentioned above from the United

States (Chow and Watson, 2002), the authors suggested that "*Acidification may be advisable when sampling particle sizes larger than 2.5 μm, when samples are acquired at locations where carbonate carbon is expected to be high, or when >800℃ temperatures are applied during thermal evolution carbon analysis*".

The above suggestion is well adapted for the study of HTP carbonaceous aerosols

(CAs). Similar to northern China, large sand dunes and deserts are widely distributed across the western HTP (Liu et al., 2005), and dust storms frequently occur in winter and spring (Wang et al., 2005;Fang et al., 2004). Thus, IC may account for a large portion of the CAs of the HTP. Unfortunately, the potential contributions of IC to TC and BC in HTP aerosols have been overlooked (Cong et al., 2015;Ming et al.,

2010;Aiken et al., 2014;Cao et al., 2010;Zhao et al., 2013;Li et al., 2016b). Additionally, IC contributions may be high because almost all of the reported data on CAs are based on the TSP content, which contains large volumes of coarse particles derived directly from MD. Therefore, TC and BC concentrations for the HTP are likely overestimated. In fact, all published papers on aerosols collected from remote areas of the HTP have

identified MD components, although none of these studies have directly discussed this issue or evaluated the effects of IC. For example, mineral aerosols and fugitive dust collected from the Lulang station of the southeastern HTP account for a large part of the TSP in spring when dust storms frequently occur (Zhao et al., 2013) (Fig. 1). In addition, the aluminum concentrations of TSP increase considerably during dust storm

events occurring at the Namco station of the central HTP, which clearly implies the





contributions of MD (Kang et al., 2015). Similarly, the dust loading derived from the $Ca^{2+}$ of the TSP samples collected at the Everest station reveal relatively constant MD levels throughout the year (Cong et al., 2015) .

Therefore, the contributions of IC to the TC and BC in the HTP aerosols must be
quantitatively evaluated. In this study, the TSP samples of two remote stations of the HTP were collected to evaluate the contributions of IC to the TC and BC. Seasonal variations in the extent of overestimations of TC and BC and possible causes were also examined. Finally, the corrected TC and BC concentrations of these two stations were determined based on previously published data (Ming et al., 2010;Cong et al., 2015).

BC deposition studies are closely related to studies of BC transport processes, lifetime and radiative forcing. BC deposition can be measured directly from historical media, such as sediments (Gustafsson and Gschwend, 1998;Han et al., 2016) and ice core (Ming et al., 2007;Ruppel et al., 2014), estimated from BC concentrations in the atmosphere (Jurado et al., 2008) or calculated using models (Zhang et al., 2015).
Compared to the process of greenhouse gas emission, the BC deposition process remains poorly defined and quantified (IPCC, 2013), especially in remote regions, such as the HTP, because of the presence of complex terrain and dynamic regimes(Bond et al., 2013;Bauer et al., 2013).

Thus far, only three studies have directly reported on BC depositions in the HTP.
One model indicated that the BC deposits of the central HTP were 9 mg $m^{-2}$ $a^{-1}$ (Zhang et al., 2015), which is roughly thirty times lower than the values found in two studies that evaluated lake cores of Namco Lake and Qinghai Lake (270-390 mg $m^{-2}$ $a^{-1}$) (Fig. 1) (Cong et al., 2013;Han et al., 2011). If the HTP BC is primarily derived from areas outside of the plateau because of long-range transport (Xu et al., 2009b;Luthi et al.,
2015) and the BC of the lake sediment only reflects atmospheric deposition as declared in the above mentioned articles based on lake cores (Cong et al., 2013;Han et al., 2011), then the above values should at least be comparable. Therefore, the discrepancies between these studies show that lake cores may overestimate BC deposition because pollutants in lake core sediments represent combined contributions from direct
atmospheric deposition and lake catchment input (Yang, 2015). Although the two





studies on lake cores mainly attribute BC to regional (long-range transport) atmospheric deposition and show an increase in BC deposition in recent years, other potential influencing factors, such as catchment inputs, were not considered or were simply overlooked; however, these factors may have contributed significantly to lake core BC

deposition.

Therefore, additional studies must be performed to provide more reliable values. For instance, other researchers have reported on BC concentration and water accumulation levels in ice cores and snowpits from the HTP (Fig. 1) (Xu et al., 2009b;Li et al., 2016c;Li et al., 2016a;Ming et al., 2008). Although these studies did not report

BC deposition patterns directly, BC deposition levels could be easily calculated out from reported data. Because glaciers are generally located at the highest altitudes of a given region, they only receive wet and dry depositions of BC from the atmosphere. Thus, BC deposits in glacial regions are assumed to closely follow atmospheric depositions. In this study, the data cited above were used to investigate the differences

in reported BC values and the potential reasons for such differences. In addition, because the HTP is situated in a remote region, BC deposition patterns in the HTP must be compared to those of other areas (e.g., the Arctic, Europe and eastern China) to better understand the patterns. Finally, actual HTP BC deposition levels were presented.

## 2. Methods

### 2.1. Collection of aerosols, surface soils and river sediments

TSP samples were collected from the Namco Station for Multisphere Observation and Research (Namco station) and the Qomolangma Station for Atmospheric and Environmental Observation and Research (Everest station) (Fig. 1) from 2014 to 2016. The Namco station is located in the center of the HTP. The Everest station is located on

the northern slopes of the Himalayas. Both of these stations are generally considered to be located in remote areas of the HTP that receive BC transported over long ranges from South Asia, and several studies on BC have been conducted in this region (Chen et al., 2015;Cong et al., 2015;Ming et al., 2010;Li et al., 2016a). TSP samples were collected using 90 mm pre-combusted (550°C, 6 hours) quartz fiber filters (Whatman

Corp) with a vacuum pump (VT 4.8, Germany). Because the pump was not equipped





with a flow meter, the air volumes passing through each filter could not be determined (Li et al., 2016d); however, this did not influence the objectives of this study (e.g., relative concentrations of TC and BC in the original and acid-treated samples). Four field blank filters were also collected from each station by exposing the filters in each sampler without pumping.

To compare the BC concentrations of the Namco Lake cores, two surface soils and four suspended particles of four rivers in the Namco Basin were collected during a period of peak river flow in 2015. A <20 μm fraction of these samples was extracted (Li et al., 2009) and treated (Han et al., 2015) to measure the BC concentrations. In addition, ten surface soil samples around the Everest station were collected to study the pH values.

## 2.2. Measurement of BC and elemental concentrations

The carbonates of the collected aerosol samples were removed by exposing a punch of samples to a vapor of 37% hydrochloric acid (HCl) for fumigation for 24 hours. Then, the treated samples were stored at 60°C for over 1 hour to remove acid left on the filter (Li et al., 2016a;Pio et al., 2007;Chen et al., 2013;Bosch et al., 2014). The OC and elemental carbon (EC, the common chemical/mass definition of BC) concentrations of both the original and treated samples were measured using the Desert Research Institute (DRI) Model 2001 Thermal/Optical Carbon Analyzer (Atmoslytic Inc., Calabasas, CA, USA) according to the IMPROVE-A protocol (Chow and Watson, 2002). The OC and BC concentrations were determined based on varying transmission signals. To evaluate the concentrations of MD, the elements of Ca, Fe, Al, and Ti in the aerosol samples were measured by inductively coupled plasma optical emission spectroscopy (ICP-OES). Reported values were subtracted from the blank concentrations. The contributions of MD (Maenhaut et al., 2002) and total carbonaceous aerosols (TCAs) (Ram et al., 2010) of the collected samples were calculated using the following equations:

$$MD = (1.41 \times Ca + 2.09 \times Fe + 1.9 \times Al + 2.15 \times Si + 1.67 \times Ti) \times 1.16 \quad (1)$$

where Si is calculated from Al assuming an average ratio of Si/Al=2.5 (Carrico et al., 2003), and



$$TCA = OC \times 1.6 + BC \qquad (2)$$

### 2.3. Adoption and calculation of BC deposition data

Previously reported BC deposition data were adopted. In addition, the BC deposition levels from the Namco station and Qinghai Lake Basin were estimated from the average BC concentrations in the atmosphere and average precipitation levels using the method described in detail in other works (Jurado et al., 2008;Fang et al., 2015) (Table. 1). In brief, the annual atmospheric deposition of BC ($\mu g\ m^{-2}\ a^{-1}$) was calculated as follows:

$$F_{BC} = F_{DD} + F_{WD} \qquad (3)$$
$$F_{DD} = 7.78 \times 10^4 \cdot V_D \cdot C_{BC\text{-}TSP} \qquad (4)$$
$$F_{WD} = 10^{-3} \cdot P_0 \cdot W_p \cdot C_{BC\text{-}TSP} \qquad (5)$$

where $F_{DD}$ and $F_{WD}$ are the seasonal dry and wet deposition ($\mu g \cdot m^{-2}$); $V_D$, $P_0$ and $W_p$ are the dry deposition velocity of aerosol ($0.15\ cm \cdot s^{-1}$), the precipitation rate (mm) of a given season and the particle washout ratio ($2.0 \times 10^5$), respectively; and $C_{BC\text{-}TSP}$ is the BC concentration of the TSPs ($\mu g \cdot m^{-3}$). The seasonal BC concentrations from the Namco station were monitored by AE-31, and the average precipitation levels at the station were recorded from 2014-2015. The BC concentrations in Qinghai Lake are reported in (Zhao et al., 2015), and the average 1961-2010 precipitation levels recorded by the China Meteorological Administration from the Huangyuan station in the lake basin were used. The values for these two areas used in the BC deposition calculations are shown in Table 1.

### 3. Results and discussion

### 3.1. Actual BC concentrations in the atmosphere over the HTP

### 3.1.1 Contribution of carbonate carbon to both TC and BC

In this study, we found that carbonate carbon contributes significantly to the BC concentrations as well as to the TC and OC concentrations of the aerosols at the Namco and Everest stations, especially during non-monsoon periods (winter and spring) when dust storms occur frequently at Namco station. The ratios of the TC, OC and BC levels of the aerosols treated with acid ($TC_A$, $OC_A$ and $BC_A$) to those of the original samples



(TC$_O$, OC$_O$ and BC$_O$) for the Namco and Everest stations were recorded as $0.81\pm0.13$, $0.78\pm0.10$ and $0.53\pm0.37$, respectively and $0.76\pm0.12$, $0.78\pm0.12$ and $0.65\pm0.26$, respectively. As proposed in previous work (Chow and Watson, 2002), BC concentrations are more heavily influenced than OC and TC concentrations because

carbonates are more prone to decomposition at high temperatures during OC and BC analyses. The OC concentrations in the treated samples used in this study also decreased, indicating that carbonates can also decompose at low temperatures (Karanasiou et al., 2010). Seasonal TC$_A$/TC$_o$ variations at the two stations differed (Fig. 2), and obvious seasonal variations of low and high TC$_A$/TC$_o$ ratios appeared during the non-monsoon

and monsoon periods for aerosols recorded at the Namco station, respectively (Fig. 2), which is consistent with heavy dust storms occurring during non-monsoon periods. However, the seasonal patterns of the TC$_A$/TC$_O$ ratio at the Everest station were not obvious, in line with relatively stable seasonal variations of Ca$^{2+}$ recorded at this station (Cong et al., 2015). Moreover, the MD/(MD+TCA) levels recorded at the Namco

station during non-monsoon periods were higher than those recorded during monsoon periods, whereas the corresponding values at the Everest station were nearly the same during both periods (Fig. 3). Compared with those of other areas, the MD/(MD+TCA) values recorded at the two stations were higher than those recorded at the NCO-P station (70% and 73% during the pre-monsoon and monsoon periods, respectively) located on

the southern slopes of the Himalayas (Decesari et al., 2010), which may be related to the serious levels of South Asia pollutants at the NCO-P station and the polluted clouds that are easily transported to this station. Alternatively, the aerosols of the NCO-P stations may carry fine particles of PM$_{10}$ with low levels of MD.

     The Everest station is located in a dry river valley with sparse vegetation cover

(a typical barren site), and MD derived from local surface soil contributes considerably to aerosols collected during monsoon periods. However, the Namco station is located in a typical grassland region with limited amounts of locally sourced dust that during monsoon periods. Additionally, the Everest station is located in the rain shadow of the Himalayas; thus, precipitation levels recorded at the Everest station (179 mm from

January 2014 to January 2015) are much lower than those recorded at the Namco station





at the same period of 302 mm, which results in high MD concentrations in the atmosphere of Everest station during monsoon periods. Potentially biased aerosol samples caused by carbonates have been proposed to occur in arid areas with alkaline soils (Chow and Watson, 2002). Because of dry weather conditions, the pH values of

soil around the Namco and Everest stations are as high as 8 (Li et al., 2008) and 8.3, respectively, implying intensive carbonate contributions. During non-monsoon periods, MD is mainly transported by westerlies from the arid western HTP, where MD is distributed across large deserts with sand dunes; thus, the aerosol samples were influenced by MD that has a high concentration of carbonates. Finally, the significantly

positive relationship (p<0.01) between Ca and IC ($TC_O$-$TC_A$) for the aerosols of these two stations further demonstrates the contributions of $CaCO_3$ to aerosol IC (Fig. 4). A similar phenomenon was also found in an aerosol study of southern Europe (Karanasiou et al., 2010), which implies that this is a common phenomenon found in arid regions.

### 3.1.2  Actual BC concentrations of two stations and implications

In summary, we clearly showed that the presence of carbonates from MD has led to overestimations of the HTP TC levels in the TSP samples by roughly 19±13% and 24±12% for the Namco and Everest stations, respectively, which were higher than the corresponding value of 10% found for coarse particles of the central Mediterranean region of Europe (Perrone et al., 2011). In addition, the related BC values were

overestimated by approximately 47±37% and 35±26%, respectively, thus implying that the actual BC concentrations at these two stations were lower than previously reported values by approximately one half and two thirds. Therefore, based on previously reported BC concentrations (Ming et al., 2010;Cong et al., 2015), the actual BC concentrations at the Namco and Everest stations were estimated at 44 ng m$^{-3}$ and 164

ng m$^{-3}$, respectively.

Although carbonates decompose at a relatively high temperature of roughly 800°C, the presence of carbonates leads to an overestimation of both BC and OC concentrations because certain components of carbonates appear in very fine particles, which leads to carbonate decomposition at relatively low temperatures. A similar phenomenon was

found via the measurement of BC levels in ambient samples in southern Europe




(Karanasiou et al., 2010). In addition, the BC and OC concentrations measured in the acid-treated samples presented several uncertainties. First, the acid-treated ambient samples transfer components of OC to BC, which leads to increased BC concentrations (Jankowski et al., 2008). However, this phenomenon was not common in the aerosol

samples examined in this study, although several samples showed higher BC concentrations in the acid-treated samples at both stations (Fig. 2). Because $BC_A$ can not be higher than $BC_o$, those samples with $BC_A/BC_O$ above one was set as one in calculation of the average value at two stations. Finally, the generally lower TC values found in the acid-treated samples clearly showed that carbonates contributed to the TC

contents in the studied aerosol samples.

The influence of carbonate carbon on TC has been observed for $PM_{2.5}$ samples from arid areas (Zhao et al., 2015;Karanasiou et al., 2010), where this phenomenon should be more obvious for TSP samples. Because dust storms of the northern and western parts of the HTP are more severe than those of the two studied stations during

the non-monsoon period, the effect of carbonates on the concentrations of OC and BC should be more pronounced in such areas and must be seriously considered in future studies. The BC values of the TSP samples in the northern HTP have also been reported (Cao et al., 2005;Ho et al., 2011), it is naturally inferred that these BC concentrations are likely overestimated by at least 45%. Correspondingly, related studies on other

issues, such as BC radiative forcing and atmospheric transport models, in the HTP based on in situ BC concentrations must be adjusted.

### 3.2. Actual BC deposition in the HTP

In general, the BC deposition levels measured via different methods should be consistent for a given region. For instance, in the seriously polluted region of eastern

China (Chen et al., 2013;Yan et al., 2015), the BC deposition level recovered from a lake core was 1,660 mg m$^{-2}$ a$^{-1}$ (Han et al., 2016), which is consistent with the values of northern China calculated from the BC concentrations in aerosols (Fang et al., 2015) and determined via in situ monitoring (Tang et al., 2014) (Table 2). Similarly, a BC deposition level of 50 mg m$^{-2}$ a$^{-1}$ was measured from a lake core in Sweden (North

Europe), and this value was similar to that of ice cores (76 mg m$^{-2}$ a$^{-1}$) drilled from



glaciers in Switzerland (West Europe) (Preunkert et al., 2000;Jenk et al., 2006).
Therefore, lake cores from polluted areas of East Asia and less polluted areas of Europe
closely reflect atmospheric BC deposition levels. For remote areas, the BC deposition
levels measured from an ice core in the Arctic and from deep ocean sediment samples

from the western South Atlantic were comparable of 27 mg m$^{-2}$ a$^{-1}$ and 10 mg m$^{-2}$ a$^{-1}$,
respectively (Lohmann et al., 2009;Ruppel et al., 2014).

### 3.2.1. Overestimated BC deposition from lake cores of the HTP

Large discrepancies were found among the reported BC deposition values in the
HTP. BC deposition levels derived from lake cores from Namco Lake (NMC09) and

Qinghai Lake were 260 mg m$^{-2}$ a$^{-1}$ and 270-390 mg m$^{-2}$ a$^{-1}$, respectively, which re much
higher than those derived from ice core and snowpit samples of the HTP (Table 2). We
proposed that the BC deposition in the lake cores of Qinghai Lake mainly reflected
atmospheric deposition followed by catchment inputs. However, the NMC09 value of
Namco Lake was mainly influenced by catchment inputs.

Lake core-derived BC deposition in Qinghai Lake was only 2-3 times higher
than that estimated from the BC concentrations of PM$_{2.5}$ in the atmosphere (Zhao et al.,
2015). Because PM$_{2.5}$ does not include all particles in the atmosphere, the actual BC
concentration in the atmosphere should be higher than that of PM$_{2.5}$; therefore, the
atmospheric BC deposition should be more similar to that of a lake core. In addition, a

previous study showed that roughly 65% and 22% of the deposition in surface
sediments of Qinghai Lake results from atmospheric deposition and catchment inputs
(Wan et al., 2012), further demonstrating the significant effects of atmospheric
deposition on lake core sediments. Therefore, if the BC concentrations in atmospheric
particles and catchment inputs are the same, then the atmospheric BC deposition

measured from the Qinghai Lake cores is overestimated by approximately 35%.

Correspondingly, catchment inputs account for a large proportion of the NMC09
samples. Because of its inert characteristics, BC is widely distributed throughout
environmental materials, such as soil and river sediments (Cornelissen et al.,
2005;Bucheli et al., 2004). Therefore, river inputs contribute to sediments as well as to

BC deposits. For instance, BC concentrations within the <20 μm fraction of surface soil





and sediment in the Namco Basin reach $0.78\pm0.48$ mg g$^{-1}$, which close to that of the Namco Lake cores of 0.74 mg g$^{-1}$. In addition, several findings have demonstrated the contributions of catchment inputs to Namco Lake cores as shown in the following sections.

First, a large glacial area (141.88 km$^2$) is distributed across the Namco Basin (Fig. 5), and large volumes of glacier meltwater and sediment pour directly into the lake (Wu et al., 2007). For example, increases in glacier meltwater in the Namco Basin triggered by increasing temperatures and precipitation account for roughly 50.6% of the lake's volume and have augmented the lake volume over the last 30 years (Zhu et al.,

2010). Originating at high-elevation glacier terminals, these rivers flow at a steep angle, and large volumes of suspended allochthonous sediments are transported into Namco Lake annually (Doberschütz et al., 2014). For instance, one study of the Zhadang glacier basin found remarkably high pollutant yields from the source river water because of the steep gradient (Sun et al., 2016). Similarly, abnormally high BC deposition levels in

Europe were also observed in lake cores of a glacier feed lake as a result of glacier meltwater effects (Bogdal et al., 2011).

    Second, previous studies on the accumulation rates of lake cores have revealed significant contributions of riverine particles. The accumulation rates of Namco Lake cores (core 08-1) are consistent with the precipitation variations recorded in the Namco

Basin over the last 60 years (Fig. 5A) (Wang et al., 2011), which indicates that heavy precipitation promotes the transportation of large riverine particles to the lake, thus increasing the accumulation rates in the lake cores. Interestingly, the mean grain size of another lake core (core NMC09) showed a significantly positive relationship with precipitation (Fig. 5B), thus reflecting the same phenomenon that catchment inputs

cause lake core accumulations (Li et al., 2014). Because the drill sites of these two lake cores are located not at the same site (Fig. 5), their similar catchment input characteristics reflect a common feature of Namco sediment. As shown above, the BC concentrations of fine fraction of river sediments are nearly equivalent to those of lake cores, so that additional catchment inputs will increase the BC deposition levels within

lake cores.



Third, the atmospheric deposition of BC calculated from BC concentrations in the atmosphere is much less significant than the deposition of BC recorded in the Namco Lake cores (Fig. 6), further reflecting the dominant contributions of catchment inputs relative to atmospheric inputs in lake cores.

Our findings indicate that variations in BC depositions in Namco Lake mainly reflected catchment inputs rather than atmospheric inputs; thus, atmospheric deposition plays a minor role relative to catchment inputs. Qinghai Lake and Namco Lake are the two largest lakes in the HTP, and they were assumed to be weakly influenced by catchment inputs. However, the above discussion shows that catchment inputs play a

dominant role in the sediment formation of these lakes. Because most lakes in the HTP have increased in area over the last 20 years (Zhang et al., 2016), this phenomenon likely occurs in many other lakes of the HTP, especially in small lakes scoured by glacier feed streams (Bogdal et al., 2011). Therefore, caution should be exercised when studying the atmospheric deposition of pollutants (e.g., BC) into the sediments of small

lakes in the HTP.

### 3.2.2. Actual atmospheric BC deposition and potential uncertainties

    BC deposits derived from ice cores and snowpits may be more similar to the actual values observed for the HTP, and this hypothesis is supported by the following findings. First, BC deposition levels in the snowpits of different glaciers are consistent. For

example, the estimated BC deposition levels of Laohugou, Tanggula, Zhadang, Demula and Yulong are 25, 21.3, 20, 14.5 and 20.2 mg m$^{-2}$ a$^{-1}$, respectively (Table 2), which reflects a homogeneous spatial distribution of BC deposition. The above values are also similar to those of ice cores described in other articles (e.g., 18 and 12 mg m$^{-2}$ a$^{-1}$ for the Muztagh Ata and Zuoqiupu glaciers, respectively (Xu et al., 2009b;Bauer et al.,

2013), and 10.1 mg m$^{-2}$ a$^{-1}$ for the East Rongbuk glacier (Ming et al., 2008) (Table 2). Second, these values are nearly equivalent to those of atmospheric BC deposits derived from completely different methods (e.g., Community Atmosphere Model version 5 (Zhang et al., 2015) and other models (Bauer et al., 2013) (Table 2). Third, the BC deposition levels of the snowpits/ice cores of the HTP are comparable to those of other

remote areas (e.g., ice cores of the Arctic (Ruppel et al., 2014) and surface sediments





of deep oceans (Lohmann et al., 2009). In summary, despite the complex topography of the HTP, atmospheric BC deposition in its glacial region presents a limited spatial discrepancy of 10-25 mg m$^{-2}$ a$^{-1}$.

Despite recent technological achievements, accurately measuring BC concentrations in ambient samples still represents a challenge in atmospheric chemistry research (Andreae and Gelencser, 2006;Bond et al., 2013). Because the methods used to measure BC concentrations and determine BC deposition levels described above are not the same, uncertainties will be introduced when directly comparing the results from different studies. For instance, different protocols used to determine temperature

increases will lead to differences in the BC concentrations obtained via different thermal-optical methods (e.g., NIOSH vs. IMPROVE) for the same sample (Karanasiou et al., 2015;Andreae and Gelencser, 2006). In general, BC concentrations derived from the IMPROVE method are 1.2-1.5 times higher than those derived from the NOISH method (Reisinger et al., 2008;Chow et al., 2009), and BC concentrations derived from

the EUSAAR_2 temperature protocol are twice as high as those derived from the NIOSH protocol (Cavalli et al., 2010). Furthermore, lake core samples must be pretreated by dissolution in HCl and HF several times, as well as centrifugation, transfer and filtration, prior to measurement via the thermal-optical method (Han et al., 2015). This complex process and the thermal-optical method produce additional uncertainties.

However, because of the complex chemical properties of ambient samples, the "best" thermal-optical protocol has not been identified (Karanasiou et al., 2015), and an exact ratio for BC produced from different methods is difficult to determine. For example, aerosols with high biomass-sourced concentrations correspond to larger differences among the BC results of different methods (Cheng et al., 2011). Therefore, in this study,

although the direct comparison of BC concentrations and deposition levels across different studies presents certain limitations, large differences of approximately thirtyfold between the lake core and snowpit data are still reliable because of relatively small differences (roughly 2 times) in the BC concentrations derived by the different methods.




### 4. Conclusions

The BC concentration and deposition levels in the HTP region, which presents the largest glacial area at middle latitudes, were investigated and reevaluated in this article. Our findings indicated that carbonate carbon contributions from MD have led to overestimations in previously reported BC concentrations of TSPs from the remote Namco and Everest stations in the central and southern HTP by approximately 47±37% and 35±26%, respectively. After omitting the contributions of carbonate carbon, the actual BC concentrations at the Namco and Everest stations should be 45 ng m$^{-3}$ and 169 ng m$^{-3}$, respectively. In addition, the levels of OC and TC in the TSPs at the Namco and Everest stations were also overestimated by 22±10% and 22±12%, 19±13% and 24±12%, respectively. Large arid areas that receive low precipitation are distributed across the western and northern HTP; thus, the effects of carbonates on BC measurements should be more significant in such areas and must be considered in future related studies. In addition, TSP samples must be treated with acid to eliminate the effects of carbonates prior to measuring BC. A comparison between the BC deposition values derived from different methods and materials showed that because of catchment inputs, the BC deposition levels derived from lake cores of the HTP were higher than the actual atmospheric deposition values. The lake cores examined in this study were drilled from the two large Qinghai and Namco Lakes, and lakes of this size should only be slightly influenced by catchment inputs. Therefore, the catchment inputs into smaller HTP lakes should be more intense, which should be considered in future studies. Correspondingly, the BC deposition values measured from snowpits and ice cores of glacial regions were similar to those obtained via models and consistent with values obtained for other remote areas around the world (e.g., Arctic and deep ocean areas); thus, they reflect the actual atmospheric deposition values of BC. Although the HTP is located adjacent to seriously polluted regions of South Asia and East China, the HTP BC deposition levels are relatively low because of the high elevation. Finally, our results indicated that the atmospheric BC deposition values of the HTP ranged from 10 m$^{-2}$ a$^{-1}$ to 25 mg m$^{-2}$ a$^{-1}$, with low and high values appearing in the central and peripheral areas of the HTP, respectively.





**Acknowledgements**

This study was financially supported by the National Nature Science Foundation of China (41675130, 41225002 and 41421061), State Key Laboratory of Cryospheric Science (SKLCS-ZZ-2008-01), and the Academy of Finland (decision number 268170). The authors gratefully acknowledge the staff at the Namco and Everest stations for their assistance with sample collection.

485

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



**Table 1** Precipitation (mm) and BC concentration (ng m$^{-3}$) values used for the

BC deposition calculations of Namco Lake and Qinghai Lake.

|  | Namco Lake | | Qinghai Lake | |
|---|---|---|---|---|
|  | precipitation | BC concentration | precipitation | BC concentration |
| Spring | 29.65 | 135.86 | 77.51 | 1000 |
| Summer | 190.05 | 90.97 | 244.02 | 530 |
| Autumn | 79.72 | 86.58 | 89.78 | 690 |
| Winter | 2.95 | 93.55 | 3.81 | 1050 |





**Table 2** Monitored or recovered BC deposits (mg m$^{-2}$ a$^{-1}$) from the HTP and other regions of the world.

| Region | Sites | Deposition | Period | References |
|---|---|---|---|---|
| Tibet | Zuoqiupu glacier | 12 | 1970-2005 | 1 |
| | Muztagh Ata | 18 | 1970-2005 | 1 |
| | East Rongbuk ice core | 10.2 | 1995-2002 | 2 |
| | Laohugou glacier | 25 | 2013-2014 | 3,4 |
| | Tanggula glacier | 21.2 | 2013-2014 | 3,4 |
| | Zhangdang glacier | 22.8 | 2013-2014 | 3,4 |
| | Demula glacier | 14.4 | 2013-2014 | 3,4 |
| | Yulong glacier | 20.3 | 2013-2014 | 3,4 |
| | Model results of central TP | 9 | 2013-2014 | 5 |
| | Namco Lake core | 260 | 1960-2009 | 6 |
| | Qinghai Lake core | 270-390 | 1770s-2011 | 7 |
| | Aerosol of Namco station | 10.5 | 2005-2007 | 8 |
| | Aerosol of Qinghai Lake | 92.7 | 2011-2012 | 9 |
| Europe | Saanajärvi lake core, Sweden | 30 | 2012 | 10 |
| | Fiescherhorn glacier, Switzerland | 76 | 1900-1940 | 11 |
| | Stora Frillingen lake core, Sweden | 50 | 1990-2000 | 12 |
| Arctic | Svalbard ice core | 27 | 1993-2003 | 13 |
| deep ocean | South Atlantic sediments | 5-78 | Surface sediments | 14 |
| East China | Chaohu lake core, East China | 1160 | 1980-2012 | 15 |
| | Northern China | 1660 | Around 2010 | 16 |
| | Northern China Plain | 1500 | 2008-2009 | 17 |

Note: 1: (Bauer et al., 2013); 2: BC concentration (20.3 ng.g$^{-1}$) and snow accumulation (500 mm) were adopted from (Ming et al., 2008) and (Li et al., 2016c), respectively; 3: (Li et al., 2016c); 4: (Li et al., 2016a); 5: (Zhang et al., 2015); 6: (Cong et al., 2013); 7: (Han et al., 2015); 8: calculation in this study; 9: calculation in this study. BC concentration was adopted from (Zhao et al., 2015); 10: (Ruppel et al., 2015); 11: BC concentration (31 ng.g$^{-1}$) and snow accumulation (2450 mm) were adopted from (Jenk et al., 2006) and (Preunkert et al., 2000), respectively; 12: (Elmquist et al., 2007); 13: (Ruppel et al., 2014); 14: (Lohmann et al., 2009); 15: (Han et al., 2016); 16: (Fang et al., 2015); 17: (Tang et al., 2014).





**Figure Captions**

**Figure 1** Selected study sites covering the HTP stations, lakes and glaciers

**Figure 2** Seasonal variations in the BC and TC concentrations in the original and acid-treated samples of aerosols measured at the Namco and Everest stations.

**Figure 3** Percentage of MD and TCA to their sum during both non-monsoon period

and monsoon period at Namco station and Everest station.

**Figure 4** Relationship between aerosol IC and $Ca^{2+}$ at the Namco and Everest stations.

**Figure 5.** Similar variations in precipitation and mass accumulation rates (A) (Wang et al., 2011) and significant relationships between mean precipitation and mean grain size (B) (Li et al., 2014) in the Namco Lake cores.

**Figure 6** Comparison of the atmospheric BC deposition levels derived from the glacial region, model, lake cores and values calculated from BC concentrations in the aerosols of the HTP.





**Figure 1**

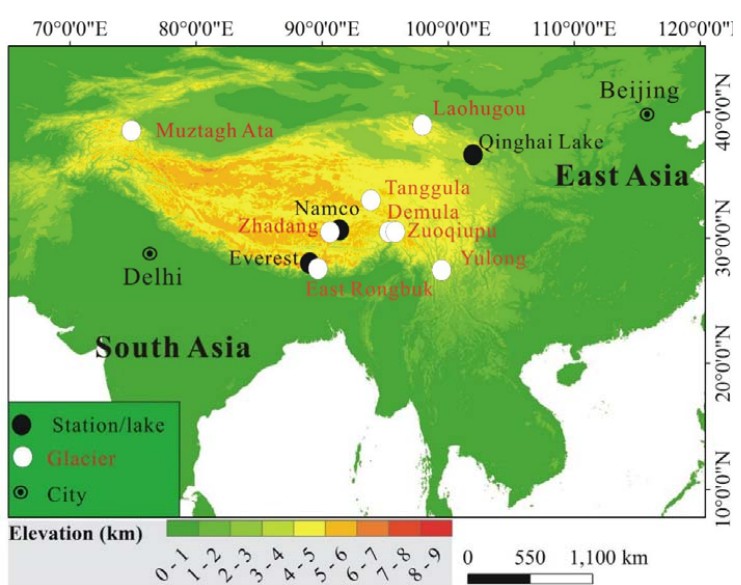




**Figure 2**

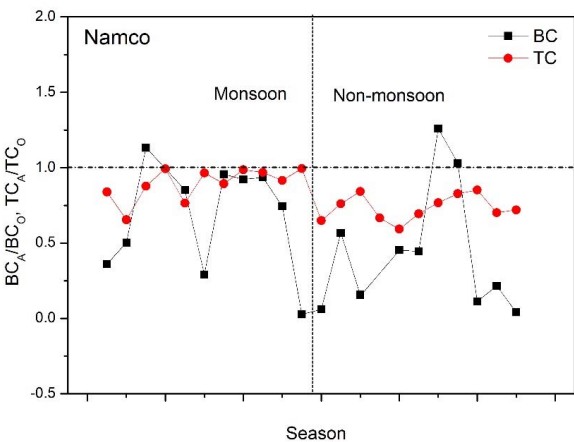

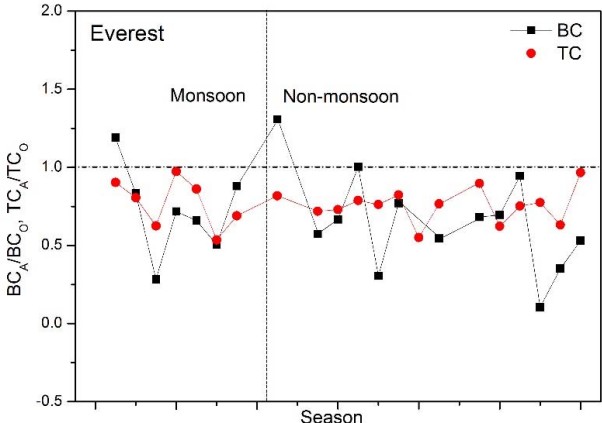




**Figure 3**

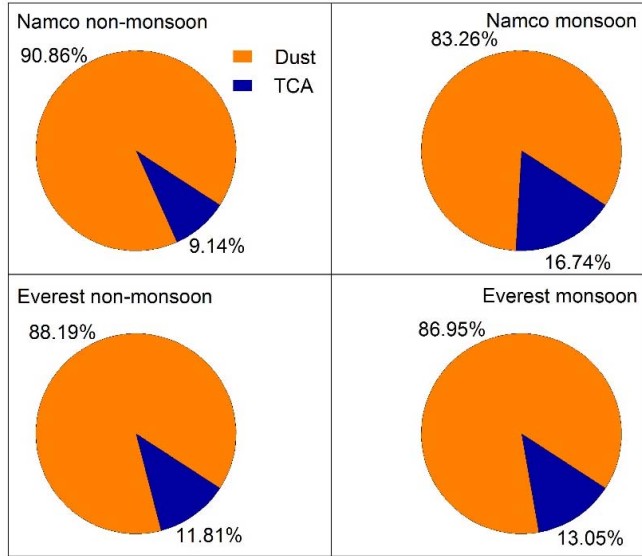



**Figure 4**

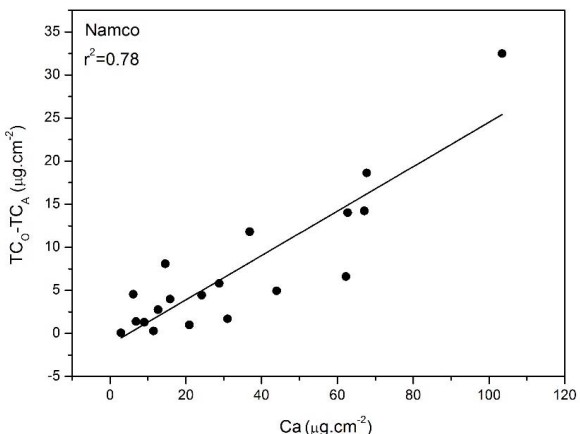


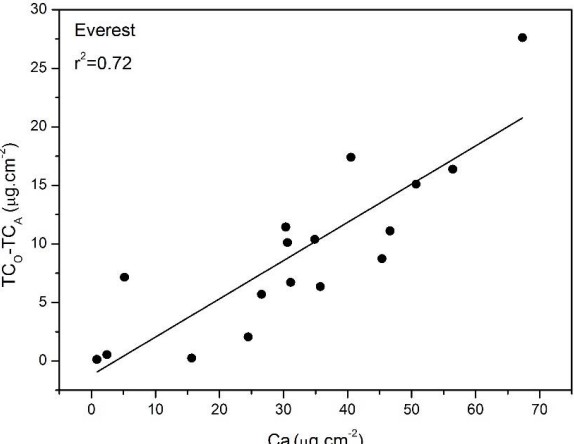





**Figure 5**

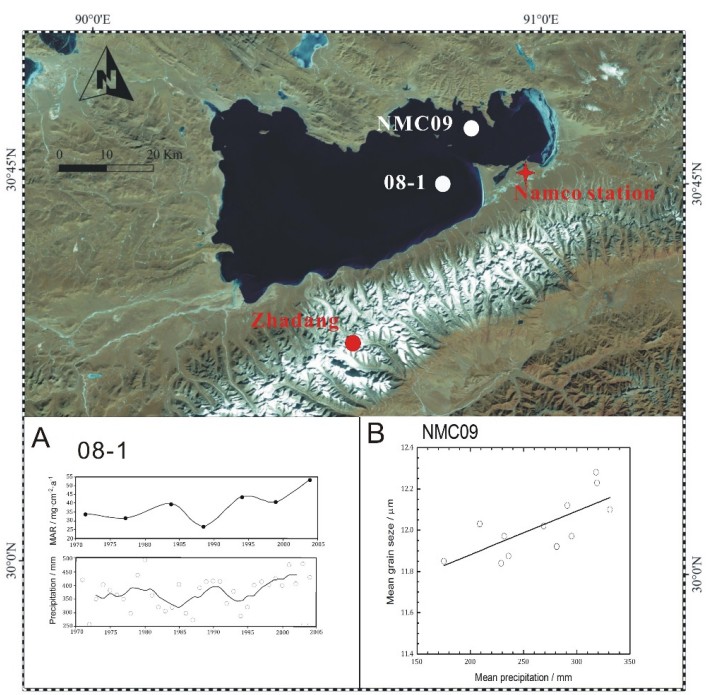






**Figure 6**

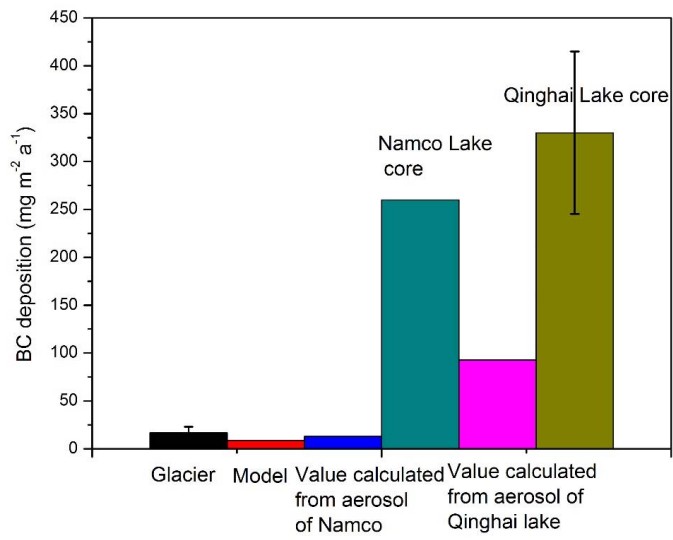