# Peer review of "Reevaluating black carbon in the Himalayas and the Tibetan Plateau: concentrations and deposition"

_Atmospheric Chemistry and Physics, 2017_

## Referee Comment (RC1) · Anonymous Referee #1 · 2 Jun 2017

Comments on Âń Reevaluating the black carbon in the Himalayan and Tibetan Plateau : concentration and deposition Âż by Chaoliu Li et al :

This paper describes a reevaluation of the black carbon content mostly at two sites in the Himalayan and the Tibetan plateau, the Everest and Namco stations. The authors consider that the BC concentrations were overestimated due to an underestimation of the carbonates from mineral dust. They also compare various BC measurement methods (in snow and glacier, in lakes and in the atmosphere) to test the coherence of deduced BC concentrations. They found that the BC concentrations measured in lake cores are mainly driven by river sediments and not atmospheric deposition. The adjusted BC concentration is then estimated as ranging from 10-25 mg m-2 a-1.

Main comments: - The introduction is very long and could be better structured to obtain

a clear view of the already published results, the problems related to the estimation of BC concentration and the analysis presented in this study. - Several species containing carbon are described (IC, CA, BC, OC, EC, TCA...) and one subject of this study concerns the wrong estimate of BC. BC is however not the only carboneous compound that absorbs -and therefore contributes to atmospheric warming- and that modify the surface albedo - and therefore modifies the glacier persistence. Can you please describe the effect of these various carboneous compounds on the climate ? - For example, you mention (§3.1.2) that the presence of carbonate led to an overestimation of the HTP TC levels. TC= total carbon. Is IC not comprised into "total carbon" ? is the effect of IC (once it is converted in equivalent BC) is air different than BC regarding the light absorption and warming of the air? Is the effect of IC different than the BC one regarding the surface albedo of glacier with deposited dust ? It is really important to have a clear discussion of these various effects to demonstrate that your main results (the BC concentration is partially due to the presence of carbonates) is important for climate analyses. - It seems (§3.2.1) that the fact that BC in lake core is largely influenced by catchment inputs was already demonstrated in several publications. Please specify clearly what is the new input of your research ! - Does the IC content depend on the mineralogical composition of MD? If yes, what are the difference between various mineral? Does it change between the various deserts around the world? - Lines300-305: at this place you mention for the first time that the acid-treated samples transfer OC to BC components. Depending of the rate of this process (that perhaps also depends on temperature), the discussion of figure 2 and §3.1.1 has to be changed. Moreover, this information is really necessary to be explained under §3.1.1 - English should really be improved! Minor comments: - Line 45: please check the language - Lines 71-73: please rephrase + are you sure that all these studies have methodological limitations bounded to the mineral dust underestimation and the impact of catchment inputs? -Line98-99: there is probably other species which concentration depends on particle size. Please mention them (see also further comment on BC content as a function of size distribution). - Lines 118-121: It is obvious that the MD content during dust storm

is high !+ Please add the reference for this results - -line 142: ... than the values measured in lake cores at Namco and Quinghai lakes... - Line 143-144: Do you mean that the HTP BC content is "measured" outside the plateau or that BC content comes from other regions that the HPT plateau ? - Line 156: which kind of values? Other measurements ? other kind of data? Other analysis and methodologies ? - Line 159-160: which quantity is then measured if they did not report BC deposition directly ? What do you mean by "deposition pattern" ? - 172: please delete (Namco station) - 204: I suppose the blank concentrations were subtracted from the measured concentrations ? - 213: "previously reported BC deposition data were adopted": what is the meaning of this sentence ? - 225-226: CBC-TSP is then the equivalent BC (usually reported as eBC) concentration in the atmosphere ?, From where do you take the dry deposition velocity and the particle washout? - 231: the values used in the BC deposition calculations for these two areas are shown... - 236: this is probably your main results. You have first to present the measurements and then to give the main conclusion. At this place, I can only say that up to now you have not given any proof for this statement. -239-246: I do not really agree with your statement "BC concentrations are more heavily influenced than OC and TC": the ratio BCa/BCois lower than for the one of OC or TC, but the uncertainties are really high (0.37 and 0.26) leading to a much more nuanced conclusion! Is it possible that the uncertainty differences between NamCo and Everest is bounded to another kind of carbonate dissociated at various temperatures ? - Figure 2: since the acid treatment is supposed to remove the IC, how is it possible to obtain values greater than 1 ? It should be really informative to add error bars for both TC and BC ratios. - 254: I suppose that you consider only MD and TCA as components of your sample, leading to MD+TCA=100%. The mention of percentage in the text (and not only in the figure) will clarify this point. Could you also give the uncertainty on dust and TCA percentages in order to estimate if the difference between non-monsoon and monsoon seasons are significant or not ? - 258: it seems to me that the NCO-P station was not mentioned before, is not in Figure 1 or in Table 2? - 262: does it mean that the NCO-P station measures PM10 and not TSP ? Was the method applied at NCO-P

СЗ

the same as at Namco and Everest station ? - 265: do you have a reference estimating the contribution of local surface soil to total MD contribution ? How is it possible to distinguish between local and desert contribution to MC ? by the carbonate types ? -267-268: please rephrase - 269-272: you compare precipitations at Namco and Everest during one year (January 2014-January 2015), that do not correspond to the unique monsoon period. I suppose that most of the precipitation occurs during monsoon, but this should be specified. - 275: you mentioned under §2 that you measured soil samples. The PH is however taken from a reference. Were the same samples used for both studies ? - 279-282 + Figure 4 : The slope of Ca versus IC is smaller (about 0.25) for Namco than for Everest station (0.30). Does it mean that there is different carbonates types at both stations ? - 289-292: here again, it is not possible to ignore the very high uncertainties in the conclusion of the impact of IC on BC concentration. - 294-292: was the BC concentrations measured by Ming and Cong estimated with a similar method that your method described in this paper ? - 294-292: please report the uncertainties of BCa/BCo to the estimated concentration in ng/m3. - 296: does the decomposition temperature of carbonate depends on the size of the particle ? If yes, you have to put a reference. - 296-300: you mixed several notion if this sentence: please clarify if you want to link the particle size distribution with the decomposition temperature, the BC and OC overestimation with the size distribution or the BC and OC overestimation with the temperature! - 308-310: it seems obvious that carbonate contribute to TC since it was stated before that it contributes to BC. - 312: where: not grammatically correct - 311 -315: could you please mention where are the station of Zhao and Karamasiou so that the reader can estimate why the dust storms are more severe at the studied stations. Please provide the same information for used stations in Cao and Ho - 319: you have to explain why you estimate the overestimation of "at least 45%" - 320-321: please see the main comment concerning the effect of carbonate or BC on climate analyses. - 318-330: if you mention the name of the stations, the reader will take much less time to understand your comparison and to find them in Table 2. - 331-336: the conclusion is too simple since you have no real correlation between measured BC in

the atmosphere and in the lake core and no proof of the polluted east asia and less polluted area in Europe! Moreover §3.2.1 shows that lake core are influence by catchment area and rivers so that you cannot here make some conclusion about Sweden lake core or about deep ocean sediment samples without taking the other results of your paper into account. - 340: re much ? - 338-344: a usual structure for a paper is not to propose an interpretation at the beginning, but to describe the results and then to draw conclusion! Do you think that you have to take into account some other parameters such as the evaporation of the lake that depends on temperature, surface and depth ? - 343: what is NMC09 ? - 347: why the BC concentration in PM2.5 should be lower than in TSP? is this statement is clear for MD due to their large size, it is not directly applicable to BC, soot being usually quite small particles. The following conclusion has therefore to be more seriously validated. - 352-355: how can you conclude that the atmosphere and the lake core should have a similar concentration of BC? This is based on 2 not proved inputs. Moreover 65% «< 100%! - Figure 5: the axes are not readable. - 407: you probably want to compare BC in ice with BC in atmosphere ? please change the sentence. - 418: the values of BC in remote areas depend on the sources and long range transport processes and cannot be compared without caution. - 423: it is perhaps better to give the discrepancy in % instead of in mass concentration - 466: if the lake are smaller, the catchment inputs are probably also smaller of the flow through the lake is larger involving a smaller deposition into the lake core. It is therefore not obvious that smaller lake with have a higher BC concentration due to catchment inputs.

---

## Referee Comment (RC2) · Anonymous Referee #2 · 9 Jun 2017

The authors address an important and underappreciated issue: the effect of mineral dust, more precisely inorganic carbonatic carbon, on black and elemental carbon measurements. The authors present some data on this, but the issue is in the current state of the manuscript not discussed thoroughly enough. Secondly, the authors discuss the role of catchment area influx of BC to lake sediment records, causing potentially flawed inferred atmospheric BC deposition results for the studied area. This second hypothesis of the manuscript is poorly justified and unnecessary for the manuscript.

In more detail, major concerns are:

- The study discusses two quite separate issues, which should be clarified notably in the introduction: First, it is discussed that (apparently mostly) atmospheric measurements of BC (more specifically EC) may be overestimated for the HTP due to mineral

dust interfering the measurements. Secondly, a quite unrelated issue of lake sediment records being affected by riverine influx of BC and not only atmospheric BC deposition. These both factors may result in over-estimations of BC or EC concentrations in these records. These two study premises should be clearly pointed out. While the hypothesis on atmospheric measurements may be justified, the authors show rash and quite poorly grounded critique on the HTP lake sediments as records of atmospheric BC deposition, as discussed below.

- The language in general is not of the required high quality (particularly in the beginning of the manuscript), and in some places even poor leading to possible misunderstandings. The word "BC deposit" is used incorrectly as a noun (e.g. line 61). It is BC causing the climate impacts, not the environmental record or matrix (i.e. "deposit", as the authors incorrectly use the term) itself.

- The statements on lines 71-74: "However, the above studies present limitations because of unique environments found in the HTP (e.g., high mineral dust (MD) concentrations in aerosols and catchment inputs to lake sediment). Therefore, the above studies should be re-investigated to better define the actual BC values." are key to the manuscript but very poorly justified by the authors. For instance, the authors cite Kaspari et al., 2011 as being one of the studies that has limitations due to e.g. high mineral dust concentrations in the HTP. However, Kaspari et al. specifically use a BC quantification method (SP2, i.e. single-particle soot photometer) which is specifically NOT influenced by dust. Furthermore, Kaspari et al. (2011) measured mineral dust separately in the same study by using iron as a proxy for mineral dust. Therefore, these statements by the authors are unjustified, and questions also the other citations by the authors. Significantly higher accuracy is required by the authors before making such bold statements.

- The authors are correct in stating (lines 78-85) that inorganic carbon (IC) may influence the total carbon (TC), particularly organic carbon (OC), and even elemental carbon (EC, thermal optical proxy for BC) concentrations. However, this applies ONLY

for OC/EC measurements with the thermal optical transmittance/reflectance (TOT/R) method. Furthermore, it depends on the used protocol for the measurements. Cavalli et al. (2010) have studied this issue thoroughly: "Natural calcite evolves in the He-mode with the EUSAAR_2 and the NIOSH protocol, and will be detected as OC, whereas it evolves in the He/O2-mode with the IMPROVE protocol and will possibly be detected as EC. Neither the NIOSH protocol nor the IMPROVE protocol address definitely this issue of CC. The NIOSH method 5040 recommends fumigation of the aerosol samples with HCl prior to thermal-optical analysis to eliminate any contribution of CC to OC and/or EC signal. However, fumigation with HCl has been shown to cause artificial loss of volatile organic acids (Chow et al., 1993) and to induce intense charring phenomena in ambient aerosol samples (Jankowski et al., 2008). The authors have not addressed these issues appropriately in the beginning of the manuscript. These issues are discussed at the very end of the manuscript which is too late, as readers familiar with these issues may have stopped reading the manuscript after this insufficient introduction.

- One of the main hypotheses of the manuscript is presented on lines 139-155. The authors claim that modelled BC deposition and observed BC deposition in e.g. lake sediments should be of comparable size. As the observed BC deposition in lake sediments is ca. 10 times higher than the modelled value, the authors claim that the discrepancy should be caused by the lake sediments not representing only BC influx from the atmosphere but also from the catchment area. This is the second main premise of the manuscript, but is unfortunately totally unjustified and false. The statement of the authors that the modelled and observed values should at least be comparable, is false. Previous studies have shown several times models to underestimate observed BC concentration and deposition values (2-5 times or even more than by a magnitude) e.g. in the Arctic and China (e.g. Koch et al., 2009; 2011; Bond et al., 2013 and references therein). These under-predictions by models often relate to difficulties in parameterizing, for instance, BC properties, aging, transportation and scavenging efficiencies properly. Moreover, the authors make a major error: observational data

is data that is used to validate modeling results that are based on estimations of BC emission strengths, and not the other way around. Modeling data is validated to be reliable based on observations, and the models are tuned accordingly. Furthermore, the authors make bold presumptions without any scientific evidence of the higher observed BC fluxes in sediment being caused by catchment influx of BC to the sediment cores. As a reviewer, I strongly suggest that the authors should consider getting familiar with basics of paleolimnology before making bold accusations about lake sediments as records of BC deposition, e.g. "Paleolimnology" by Andrew S. Cohen (2003, Oxford University Press). Basically, there are lakes that are well suitable as BC repositories as they mainly collect atmospherically deposited material. Crucial is, where the sediment core is collected, at the deepest point of the lake and preferably from a lake with very smooth bathymetry. In such a case the sediment core is not at all affected by redistributed sediment e.g. from the catchment area. So the coring site selection is crucial. Additionally, the lake sediments are dated based on lead210 deposition. As the amount of lead 210 is known for the present, and its halflife time is known, lead210 measurements vertically in a sediment core present a robust dating technique. Subsequently, the amount of lead210 in the sediment samples will also inform the researcher whether the coring site is affected by sediment redistribution or not. If not, then the amount of lead210 is the same in the surface sediment sample as in the atmosphere. If there is less or more, this is an indication of the coring spot losing or gaining additional sediment and thereby also BC (e.g. Blais & Kalff, 1995). This can be studied by the sediment focusing factor of the sediment core (Blais & Kalff, 1995). So have the authors checked what the sediment focusing factor of the studied HTP lakes is? According to their suggestion of the lakes recording 30 times too high BC deposition flux values compared to the modelled values due to sediment influx from the catchment area would mean that the sediment cores would have to receive 30 times more sediment influx from the catchment area than what they accumulate naturally. Such values are unheard of in these types of research. Consequently, the authors need to get a basic understanding of paleolimnology and lake sediments as records of contaminants before making any such suggestions. Basically, this unfounded premise greatly undermines the half of the manuscript discussing this issue.

- On lines 156-168 the authors present that other studies made based on ice core records show lower BC concentrations and deposition values than the lake sediment and that ice cores are more suitable than lake sediments to record only atmospherically deposited BC. Yes, ice cores record more reliably only atmospherically deposited BC, but when taking into account the sediment focusing factor, input of BC to the sediment cores from the catchment area can be eliminated and values more reliably presenting atmospheric deposition can be achieved (Blais & Kalff, 1995). Secondly, the authors make a mistake in comparing directly lake sediment and ice core BC records. This is because BC has been mostly analyzed with different analytical methods from these archives resulting in different types of BC particles being quantified (e.g. Hammes et al., 2007). Subsequently, even from same samples considerably different BC concentrations can be detected (e.g. Watson et al., 2005). Sure, many HTP ice core and lake sediment records measure BC as elemental carbon with thermal-optical methods from the samples so that the values should basically be comparable. However, the thermal-optical measurements of the sediments undergo extensive chemical pretreatment before the actual BC analysis and this may partly result in different BC particles being quantified. The comparison is not as straight-forward as the authors try to convey. Please, also remember to consider the different protocols in the TO-measurements as the authors have pointed out themselves. This issue is, again, discussed at the very end of the manuscript, which is not a proper structure for this.

- "Because glaciers are generally located at the highest altitudes of a given region, they only receive wet and dry depositions of BC from the atmosphere." This is a very confusing statement (line 161-162) that needs clarification. Low-elevation glaciers can be found around the world. The key is to collect the ice core from the accumulation area of the glacier, not e.g. ablation zone.

- I disagree with the comment by the authors (lines 165-168) that "In addition, because

the HTP is situated in a remote region, BC deposition patterns in the HTP must be compared to those of other areas (e.g., the Arctic, Europe and eastern China) to better understand the patterns." Europe and especially the Arctic have different sources and deposition processes for BC and comparison of HTP BC deposition with Arctic BC deposition seems unjustified.

- Line 204-205, I hope blank values were subtracted from reported values and not the other way around as suggested in the text.

- The tests of the authors to remove carbonates from the atmospheric sample filters by fumigation are valuable and interesting. However, most useful results could have been achieved if these measurements were conducted using all the appropriate different temperature programs available (e.g. NIOSH, IMPROVE and EUSAAR_2), because all these have been used in literature, and carbonatic carbon evolves at different analysis stages during these protocols.

- The discussion on lines 301-310 indicates that the authors aren't really sure what happens to carbonates after the acid fumigation. Does carbonate affect the OC or the EC quantification? Seemingly both, and varyingly from sample to sample. To me, this causes high uncertainties in the interpretations of the data. This procedure: "samples with BCA/BCO above one was set as one in calculation of the average value at two stations" sounds quite artificial and there is a lot of room for intense charring being caused by the fumigation (Jankowski et al., 2008) obstructing the kind of analysis the authors are attempting.

- The statement on lines 323-324 "In general, the BC deposition levels measured via different methods should be consistent for a given region." is false for even within one environmental matrix, e.g. sediments (Watson et al., 2005; Hammes et al., 2007; Han et al., 2011) or snow samples (e.g. Lim et al., 2014) let alone between different environmental records (e.g. Rose & Ruppel, 2015). It is absolutely inappropriate to compare BC deposition values measured using different analytical methods as these

measure different BC particle types, as the previously cited work clearly reveals. By comparing suitable selected work with one another on lines 324-336 the authors were able to compare some similar values recorded with different methods for some regions, but at the same time neglected a huge amount of data available which did not suit this statement.

- Section "3.2.1. Overestimated BC deposition from lake cores of the HTP" contains a lot more promising discussion that expected on the false presumptions presented in the introduction. This section could be clarified and made more convincing with inclusion of the sediment focusing factor calculation and discussion from the respective lakes. However, it is unclear why the authors want to compare in Table 2 BC deposition results from all around the world to HTP values? It's inappropriate and it seems that the authors have simply selected sites that support their points and are not presenting all available data. For instance, all Greenland ice core data is neglected, although there is a lot of data available, but it just happens to show almost a magnitude lower BC deposition (e.g. McConnell 2010) than other Arctic results, for which reason it seems not to have been presented here. I strongly suggest leaving away the discussion and data of other areas that HTP and Asia. Particular attention should be paid to the fact with how different methods the results in the different articles are produced.

- Also in Section 3.2.2., please remove any comparison of HTP values to e.g. Arctic or European BC deposition values. Such comparisons are inappropriate due to different methodologies and very different sources and BC deposition processes in these areas. Furthermore, these comparisons are not necessary for this manuscript and don't lift its significance in any way.

- Surprisingly then, the last paragraph of 3.2.2. discusses the possible uncertainties in comparing the BC deposition results between different methods and environmental archives. Good! Unfortunately, this discussion comes too late in the manuscript and is not thorough enough. Things mentioned earlier in this review should be carefully considered.

All in all, the manuscript contains some important perspectives and a little new data on carbonatic carbon in atmospheric BC samples, but it is questionable whether this is enough data to constitute sufficient scientific novelty for a publication. The amount of carbonatic carbon is measured in some atmospheric samples, but it is not estimated whether or how this would have affected e.g. previous snow and ice core measurements in the HTP, where dust is frequently present. This discussion is majorly hampered by the fact that dust, and therefore carbonate, concentrations vary substantially from sample to the next and will have variable influence accordingly. Much more discussion on these issues is required for the manuscript and even after that, it's maybe enough to publish only as a technical note.

The discussion on BC deposition values reported for the HTP in lake sediments not reliably representing atmospheric BC deposition, and that the BC concentrations and deposition should be re-evaluated in the HTP, is unjustified and poorly researched. The fact that two lake sediment records show different deposition values than for instance ice cores and atmospheric observations is ultimately not surprising. If the authors wish to further study this issue they should first try to deeply understand the factors controlling BC deposition in lake sediments and the importance of different analytical methodologies for the interpretations. All discussion on other than HTP, or Asian, BC deposition results should be removed from the manuscript. In my personal opinion, this lake sediment discussion should be completely removed from the manuscript. It is unrelated to the carbonate issue, and there is so much other, e.g. ice core, BC deposition data available from HTP, that it is unnecessary to try to re-evaluate the whole HTP BC deposition patterns based on these two sediment records showing different results than other records in the larger area.

Consequently, the authors present too little own new data and their discussion on literature data is mostly inadequate and not presented in a clear structure in the manuscript.

References:

Blais, J. M; Kalff, J. (1995) The influence of lake morphometry on sediment focussing. Limnol. Oceanogr., 40 (3), 582-588.

Bond, T. C., et al. (2013) Bounding the role of black carbon in the climate system: A scientific assessment. J. Geophys. Res.-Atmos., 188, 5380–5552.

Cavalli, F., et al. (2010) Toward a standardised thermal-optical protocol for measuring atmospheric organic and elemental carbon: the EUSAAR protocol. Atmos. Meas. Tech., 3, 79–89, doi:10.5194/amt-3-79-2010.

Chow, J. C., et al. (1993) The DRI Thermal/Optical Reflectance carbon analysis system: Description, evaluation, and applications in U.S. air quality studies, Atmos. Environ., 27A, 1185–1201.

Hammes, K., et al. (2007) Comparison of quantification methods to measure fire-derived (black/elemental) carbon in soils and sediments using reference materials from soil, water, sediment and the atmosphere. Glob. Biogeochem. Cycles, 21, GB3016, doi: 10.1029/2006GB002914.

Han, Y. M.; et al. (2011) Comparison of elemental carbon in lake sediments measured by three different methods and 150-year pollution history in Eastern China. Environ. Sci. Technol., 45, 5287-5293.

Jankowski, N., et al. (2008) Comparison of methods for the quantification of carbonate carbon in atmospheric PM10 aerosol samples, Atmos. Environ., 42, 8055–8064.

Kaspari, S. D., et al. (2011) Recent increase in black carbon concentrations from a Mt. Everest ice core spanning 1860-2000 AD. Geophys. Res. Lett., 38, L04703, doi:10.1029/2010GL046096.

Koch, D., et al. (2009) Evaluation of black carbon estimations in global aerosol models. Atmos. Chem. Phys., 9, 9001–9026.

Koch, D., et al. (2011) Coupled aerosol-chemistry-climate twentieth century transient

model investigation: Trends in short-lived species and climate responses. J. Climate, 24, 2693–2714, doi:10.1175/2011JCLI3582.1.

Lim, S. et al. (2014) Refractory black carbon mass concentrations in snow and ice: method evaluation and inter-comparison with elemental carbon measurement, Atmos. Meas. Tech., 7, 3307–3324. McConnell, J. R. (2010) New Directions: Historical black carbon and other ice core aerosol records in the Arctic for GCM evaluation. Atmos. Environ., 44, 2665–2666.

Rose, N. L., Ruppel, M. (2015) Environmental archives of contaminant particles, Blais JM, Rosen MR, Smol JP (eds.), Environmental Contaminants. Developments in Paleoenvironmental Research 18, Springer, Dordrecht, pp. 187–221.

Watson, J. G., et al. (2005) Summary of organic and elemental carbon/black carbon analysis methods and intercomparisons. Aerosol Air Qual. Res., 5, 65–102.
* * *

---

## Author Comment (AC1) · 22 Jul 2017

Dear reviewer:

  We are grateful for your constructive suggestions and questions, which greatly improve this MS. Despite some critical comments from both reviewers, you give us the overall positive assessments. You provided not only detailed overview but also lots of detailed and important questions and suggestions to the MS. Meanwhile, we also modified some mistakes that found by ourselves during the modification. In addition, the English of the MS has been improved by the professional English editor of Springer nature (Receipt code: GOTRE-F49-0710225832). According to suggestion of English editor, the title of the MS was modified to "*Reevaluating black carbon in the Himalayas and the Tibetan Plateau: concentrations and deposition*" and all the English of the MS was greatly improved.

  We show our great thanks to all the questions and suggestions and have answered all of them. Our answers and modifications in the revised MS were marked in blue. The sentence added in the revised MS was marked in red and italic.

  Because new experiment has been conducted by help of other researchers from Shandong University, three more researchers were invited as co-authors of this MS.

  If you have more questions or suggestions please let us know.

  Best wishes!

Chaoliu Li on behalf of all the co-authors

2017/7/20

**Point-by-point response to reviewer's comments**
**Anonymous Referee #1**

Comments on ´n Reevaluating the black carbon in the Himalayan and Tibetan Plateau: concentration and deposition Â˙z by Chaoliu Li et al :This paper describes a reevaluation of the black carbon content mostly at two sites in the Himalayan and the Tibetan plateau, the Everest and Nam Co stations. The authors consider that the BC concentrations were overestimated due to an underestimation of the carbonates from mineral dust. They also compare various BC measurement methods (in snow and glacier, in lakes and in the atmosphere) to test the coherence of deduced BC concentrations. They found that the BC concentrations measured in lake cores are mainly driven by river sediments and not atmospheric deposition. The adjusted BC concentration is then estimated as ranging from 10-25 mg m-2 a-1.
Thanks a lot for the detailed overview of the main points of the MS.
Main comments: The introduction is very long and could be better structured to obtain a clear view of the already published results, the problems related to the estimation of BC concentration and the analysis presented in this study.
Answer: Thanks a lot for the suggestion. The other reviewer also show the similar

feeling of the introduction. The introduction has been cut to short and those not very related the topic of the MS were deleted in the revised MS.

- Several species containing carbon are described (IC, CA, BC, OC, EC, TCA: : :) and one subject of this study concerns the wrong estimate of BC. BC is however not the only carboneous compound that absorbs -and therefore contributes to atmospheric warming- and that modify the surface albedo – and therefore modifies the glacier persistence. Can you please describe the effect of these various carboneous compounds on the climate ?

Answer: Thanks a lot for this question on the different climate forcing effects of different components of the carbonaceous matter. These expressions were double checked in the MS. It need to point out that "EC" is the common chemical/mass definition of "BC". CA contains BC and OC. Therefore, we added the climate forcing of OC because light absorption of BC has been shown in the MS. Climate forcing of IC discussed in the following question. Meanwhile, replaced "TC" by "CA" in the revised MS and related figures. The sentence "*Organic carbon (OC) is generally considered to scatter sunlight. However, some components of OC also absorb sunlight and warm the atmosphere (Andreae and Gelencser, 2006). Therefore, the contributions of IC to the OC and BC values in the HTP aerosols must be quantitatively evaluated.*" was added into the revised MS.

- For example, you mention (§3.1.2) that the presence of carbonate led to an overestimation of the HTP TC levels. TC= total carbon. Is IC not comprised into "total carbon" ?

Answer: Thanks a lot for the question and sorry of confused expression. "total carbon" contains IC and we changed the description in the revised MS. To make it clear, that sentence was modified to "*IC can also emit $CO_2$ in response to increasing temperature during measurements, thus causing an overestimation of the total carbon (TC) in carbonaceous aerosols (CAs) (Karanasiou et al., 2010)*".

Is the effect of IC (once it is converted in equivalent BC) is air different than BC regarding the light absorption and warming of the air?   Is the effect of IC different than the BC one regarding the surface albedo of glacier with deposited dust ? It is really important to have a clear discussion of these various effects to demonstrate that your main results (the BC concentration is partially due to the presence of carbonates) is important for climate analyses.

Answer: Thanks for the question. It is better to compare albedo reduction caused by mineral dust (including IC) and BC because both of them belong to light absorption particles. Previous study has shown that BC had higher light absorption ability than that of mineral dust in atmosphere. For instance, Dust particles are more weakly absorbing per mass (about 0.009 $m^2$ $g^{-1}$ at 550nm for Asian dust) (Clarke et al., 2004). The corresponding value for BC is 5 $m^2$ $g^{-1}$ at 550 nm (Bond and Bergstrom, 2006). Therefore, BC has higher albedo reduction ability than that of mineral dust at the surface of glacier (Qu et al., 2014). A sentence "*Because MD has lower influences on light than BC in the atmosphere (Bond and Bergstrom, 2006; Clarke et al., 2004) and on glacier surfaces (Qu et al., 2014), considering IC as BC will overestimate the BC-driven climate forcing.*" was added into the revised MS.

- It seems (§3.2.1) that the fact that BC in lake core is largely influenced by catchment inputs was already demonstrated in several publications. Please specify clearly what is the new input of your research !

Answer: Thanks a lot for the suggestion. A sentence "*although the influence of sediment focusing on BC deposition in lake cores has been noted in other areas (Blais and Kalff, 1995; Yang, 2015), it has not been pointed out and evaluated in the HTP.*" was added into the revised MS and some not important sentences were deleted.

Does the IC content depend on the mineralogical composition of MD? If yes, what are the difference between various minerals? Does it change between the various deserts around the world ?

Answer: Yes, IC content of different minerals are different, because IC is mainly contained in carbonate minerals. Spatially, the dryer the desert, the highest concentration of IC. But for the HTP, the mineral dust is transported mainly from the local sources of itself (Li et al., 2012). Because aerosols are well mixed when transported in atmosphere (Sun et al., 2007), the IC concentration of collected atmospheric particles should be varies little.

- Lines300-305: at this place you mention for the first time that the acid-treated samples transfer OC to BC components. Depending of the rate of this process (that perhaps also depends on temperature), the discussion of figure 2 and §3.1.1 has to be changed. Moreover, this information is really necessary to be explained under §3.1.1

Answer: Thanks a lot for the suggestion, and the other reviewer has pointed out the similar suggestion. The uncertainty of this method has been moved ahead and related explanations have been modified in the revised MS.

- English should really be improved!

Answer: The whole MS has been improved by the professional English editor of Springer nature (Receipt code: GOTRE-F49-0710225832).

  Minor comments: - Line 45: please check the language

Answer: The sentence was changed to "*Correspondingly, BC deposition derived from snowpits and ice cores agreed well with those derived from models, implying that the BC depositions of these two methods reflect the actual values in the HTP.*".

- Lines 71-73:

please rephrase + are you sure that all these studies have methodological limitations bounded to the mineral dust underestimation and the impact of catchment inputs? –

Answer: Thanks a lot for the question, which was also pointed out by the other reviewer. Sorry that there are inconsistences between two sentences and incorrect references adopted between lines 67-73. Therefore, all of those references not related to aerosol of the HTP were deleted. To make the total expression consistent, the sentence was modified to "*numerous studies have been conducted on the BC concentrations in the atmosphere (Cong et al., 2015; Marinoni et al., 2010; Ming et al., 2010; Wan et al., 2015; Zhao et al., 2013) and atmospheric BC deposition as determined from lake core sediments (Cong et al., 2013; Han et al., 2015). However, all of these studies exhibit limitations because of certain special environmental factors in the HTP (e.g., high concentrations of mineral dust (MD) in aerosols and catchment inputs to lake core sediment).*" In the revised version.

Line98-99: there is probably other species which concentration depends on particle size. Please mention them (see also further comment on BC content as a function of size distribution).

Answer: Yes, other species such as carbonates and related major elements (e.g., Ca, Fe and Mg) are closely connected to MD and particle size distribution. Because we mainly discuss IC in this MS, that sentence was modified to "*Because TSP samples contain more MD and carbonates than PM$_{2.5}$, they should have higher concentrations of IC*".

Lines 118-121: It is obvious that the MD content during dust storm is high !+ Please add the reference for this results

Answer: Thanks for the suggestion. Because this part is not very important in the introduction and the reviewer suggest that we cut the introduction. This part was cut in the revised MS. Therefore, the related reference was not added.

-line 142: : : : than the values measured in lake cores at Nam Co and Qinghai lakes: : :

Answer: Suggestions accepted and that part was modified in the revised MS.

- Line 143-144: Do you mean that the HTP BC content is "measured" outside the plateau or that BC content comes from other regions that the HPT plateau ?

Answer: Thanks for the question. The cut the introduction, this part was deleted from the revised MS.

- Line 156: which kind of values? Other measurements? other kind of data? Other analysis and methodologies ?

Answer: Here reliable values is BC deposition value, which was added into the revised MS.

- Line 159-160: which quantity is then measured if they did not report BC deposition directly ?

Answer: Sorry for the confused expression. The mean here is BC deposition can be "calculated out" not "measured". Therefore, this sentence was modified to "*in those articles*" was added to the end of that sentence.

What do you mean by "deposition pattern" ?

Answer: Thanks a lot for the question. "patterns" is an extra word and need to be deleted. In addition, because the other reviewer thought that part of comparison between the HTP and other regions was useless, that part was totally deleted from the revised MS.

- 172: please delete (Nam Co station)

Answer: "Nam Co station" deleted.

- 204: I suppose the blank concentrations were subtracted from the measured concentrations?

Answer: Yes, your assumption is what described. Therefore, a sentence "All the reported values in this study were corrected based on the values of the blanks." was added into the revised MS.

- 213: "previously reported BC deposition data were adopted": what is the meaning of this sentence ?

Answer: Sorry for the confused description. This sentence was modified to "*To determine the actual BC deposition in the HTP, previously reported data were compiled and evaluated (Table 1)*".

- 225-226: CBC-TSP is then the equivalent BC (usually reported as eBC) concentration

in the atmosphere ?, From where do you take the dry deposition velocity and the particle washout?

Answer: Thanks a lot for the question. $C_{BC-TSP}$ is the abbreviation of BC concentration of the total suspended particle (TSP) for a given region, not equivalent BC. The values of dry deposition velocity and the particle washout were taken from reference of (Fang et al., 2015), which was added in the revised MS.

- 231: the values used in the BC deposition calculations for these two areas are shown: : :
Answer: This sentence was modified according to suggestion.

- 236: this is probably your main results. You have first to present the measurements and then to give the main conclusion. At this place, I can only say that up to now you have not given any proof for this statement.

Answer: Sorry of not good expression. A sentence "*after compared with BC and OC concentrations of original and acid treated TSP samples,*" was added in front of results.

- 239-246: I do not really agree with your statement "BC concentrations are more heavily influenced than OC and TC": the ratio BCa/BCo is lower than for the one of OC or TC, but the uncertainties are really high (0.37 and 0.26) leading to a much more nuanced conclusion! Is it possible that the uncertainty differences between Nam Co and Everest is bounded to another kind of carbonate dissociated at various temperatures ?

Answer: Thanks for the suggestion. This sentence was deleted from the revised MS. We think the uncertainties were because of the different mineral dust concentrations in different aerosol samples at two stations, which was added to the revised MS.

    In addition, the other reviewer think those samples with BCa/BCo higher than 1 cannot be simply considered as 1. Therefore, the exact ratio were recalculated used the new database.

- Figure 2: since the acid treatment is supposed to remove the IC, how is it possible to obtain values greater than 1 ? It should be really informative to add error bars for both TC and BC ratios.

Answer: Thanks a lot for the suggestion. The other reviewer also pointed out this question. The main reason of the ratios higher than 1 at both stations is that sometimes the treatment of ambient samples by acid make some components of OC easily be pyrolyzed and transferred to BC (Jankowski et al., 2008). However, so far, no perfect method can deal with this problem. Therefore, those samples with ratios above 1 were deleted and the final results were recalculated. Meanwhile, a sentence "*Nevertheless, the ratio of $BC_A/BC_O$ was considered to be slightly overestimated as some portion of OC was considered BC in the acid-treated samples (Jankowski et al., 2008)*" was added in the revised MS to point out the uncertainties. Because each point of Fig. 2 represented only one sample, so that no error bar was shown on the figure. Furthermore, to make the figure more clear. The ratios for TC and BC during monsoon period and non-monsoon period were added on Fig. 2.

- 254: I suppose that you consider only MD and TCA as components of your sample, leading to MD+TCA=100%. The mention of percentage in the text (and not only in the figure) will clarify this point. Could you also give the uncertainty on dust and TCA percentages in order to estimate if the difference between non-monsoon and monsoon seasons are significant or not ?

Answer: Thanks a lot for the suggestion. "TCA" should be "CA" in the revised MS. The samples contain other components other than MD and CA. Here we study only the relative ratio of MD and CA. To make it clear, a sentence "*To evaluate the relative ratio of MD and CA, MD/(MD+ CA) values were calculated (Fig. 3)*" was added in the revised MS. Difference of MD/(MD+ CA) between non-monsoon and monsoon seasons of samples of two stations were checked. The results showed that the difference at Nam Co station was significant ($p<0.01$), but at Everest station was insignificant ($p>0.05$), which was also added into the revised MS.

- 258: it seems to me that the NCO-P station was not mentioned before, is not in Figure 1 or in Table 2 ?

Answer: Sorry that location of NCO-P station is not mentioned in both of them. Therefore, location of this station (27.95°N, 86.82°E, 5079 m a.s.l) was added in the revised article when it first appeared.

- 262: does it mean that the NCO-P station measures PM10 and not TSP ? Was the method applied at NCO-P the same as at Nam Co and Everest station ?

Answer: Yes, that research only reported data of $PM_{10}$ and $PM_1$. Therefore, data of $PM_{10}$ was adopted for the comparison with this study.

The methods used in that study was different from this MS. For instance, Ca and Mg concentrations of that study were only water-insoluble fraction, which will cause underestimate of mineral dust (Decesari et al., 2010). Meanwhile, EUSAAR_2 temperature program of Thermo-Optical Transmission (TOT) method was used in that study for OC and EC measurement, which was also different from method in this article. Therefore, a sentence "*because the measured particle size ($PM_{10}$) and the measurement methods of Ca, Mg and EC at the NCO-P station differed from those in this study (Decesari et al., 2010), uncertainties exist in such a direct comparison*" was added into the revised MS.

- 265: do you have a reference estimating the contribution of local surface soil to total MD contribution? How is it possible to distinguish between local and desert contribution to MC ? by the carbonate types ?

Answer: Yes, according a previous study at this station, even those fine particle in the atmosphere was derived mainly from local source soil (Liu et al., 2017). This reference has been added into the revised MS. Rare earth element ratios can be potential indexes to distinguish local and desert sources. For instance, in our previous research we found MD of the Tibetan soil itself is the main sources of particles loaded at glacier of the Tibetan Plateau by these indexes (Li et al., 2012). Sorry that we do not find related knowledge on how to use carbonate types to distinguish MD sources.

-267-268: please rephrase - 269-272: you compare precipitations at Nam Co and Everest during one year (January 2014-January 2015), that do not correspond to the unique monsoon period. I suppose that most of the precipitation occurs during monsoon, but this should be specified.

Answer: Thanks a lot for the suggestion. The precipitation of two station were modified to that of monsoon period in the revised MS. Accordingly, the sentence was modified to "*the precipitation level recorded at the Everest station (172 mm during the monsoon period between 2014 and 2015) is much lower than that of the Nam Co station (258*

*mm), causing high MD concentrations in the atmosphere of the Everest station during that period*".

- 275: you mentioned under §2 that you measured soil samples. The PH is however taken from a reference. Were the same samples used for both studies?

Answer: Not the same samples for two stations. Data of one study at Nam Co region is taken from our previous article (Li et al., 2008), therefore the reference was adopted. Correspondingly, pH of soil sample collected around Everest station was newly measured, which was mentioned in method part (last paragraph of the 2.1).

- 279-282 + Figure 4 : The slope of Ca versus IC is smaller (about 0.25) for Nam Co than for Everest station (0.30). Does it mean that there is different carbonates types at both stations ?

Answer: Yes, you assumption is correct and the corresponding explanation was added into the revised MS. Therefore, a sentence "*The ratio of Ca/IC was higher in the Everest station samples than that of Nam Co station, possibly reflecting different types of carbonate at these two stations.*" was added into the revised MS.

  - 289-292: here again, it is not possible to ignore the very high uncertainties in the conclusion of the impact of IC on BC concentration.

Answer: The exist of uncertainties for this conclusion was added into the revised MS. Therefore, a sentence "*Moreover, because of the large variations in the above values, the corrected BC concentrations at the two stations have large uncertainties.*" was added into the revised MS.

- 294-292: was the BC concentrations measured by Ming and Cong estimated with a similar method that your method described in this paper ?

Answer: Yes, this MS used the same method (Thermal-Optical method, IMPROVE-A) with those of two studies, which was added in the revised MS.

- 294-292: please report the uncertainties of BCa/BCo to the estimated concentration in ng/m3.

Answer: Thanks for the question. The RSD of BC concentrations of the two stations were shown based on that of BCa/BCo.

  - 296: does the decomposition temperature of carbonate depends on the size of the particle ? If yes, you have to put a reference.

Answer: Related studies were double checked and did not find the direct proof. Therefore, this sentence was deleted in the revised MS. Nevertheless, previous study showed that IC of carbonates could influence both OC and BC of aerosols (Karanasiou et al., 2015).

Therefore, this sentence was modified to "*The OC concentrations in the treated samples used in this study also decreased, indicating that carbonates can also decompose at low temperatures (Karanasiou et al., 2010)*".

- 296-300: you mixed several notion if this sentence: please clarify if you want to link the particle size distribution with the decomposition temperature, the BC and OC overestimation with the size distribution or the BC and OC overestimation with the temperature!

Answer: Thanks for the suggestion. This sentence has been rewritten and the part on fine particle size was deleted. The new sentence has been shown in the answer to the

above question.

- 308-310: it seems obvious that carbonate contribute to TC since it was stated before that it contributes to BC.

Answer: Yes, it is just repeat the issues have been pointed out in the MS. Therefore, this sentence was deleted in the revised MS.

- 312: where: not grammatically correct

Answer: Thanks a lot for the suggestion. That sentence was changed to "*Since the influence of carbonate carbon on TC has been observed in PM$_{2.5}$ samples from Qinghai Lake, Northwest China (Zhao et al., 2015), this phenomenon should be clear in the TSP samples in this study.*".

- 311 -315: could you please mention where are the station of Zhao and Karamasiou so that the reader can estimate why the dust storms are more severe at the studied stations. Please provide the same information for used stations in Cao and Ho

Answer: Study of Zhao et al., (2015) was conducted at Qinghai Lake, West China, where is near to numbers of deserts. Study of Cao et al., (2005) was conducted at city Xi'an, Middle west China, where dust storm happened frequently during Spring. Study of Ho et al., (2011) was conducted at Tongyu, Northeast China and focused mainly on dust storm events. Study of Karamasiou et al., (2015) is a review that discussed the influence of carbonate on OC and BC, so that no station can be pointed out and this reference was deleted here.

Therefore, the locations of the first three stations were added into the revised MS.

- 319: you have to explain why you estimate the overestimation of "at least 45%"

Answer: Sorry of not expressing it clear and making a mistake. As our newly calculated value, the correct expression should be 52%, which was the newly calculated value. Therefore, that sentence was modified and the reason of 52% was added in the revised MS.

- 320-321: please see the main comment concerning the effect of carbonate or BC on climate analyses.

Answer: Thanks a lot for the suggestions. The reply has been shown in the answer to previous question. The relative climate forcing has been added in the revised MS.

- 318-330: if you mention the name of the stations, the reader will take much less time to understand your comparison and to find them in Table 2.

Answer: Thanks a lot for the suggestion and the station names have been added at related positions to make it clearer.

- 331-336: the conclusion is too simple since you have no real correlation between measured BC in the atmosphere and in the lake core and no proof of the polluted east asia and less polluted area in Europe! Moreover §3.2.1 shows that lake core are influence by catchment area and rivers so that you cannot here make some conclusion about Sweden lake core or about deep ocean sediment samples without taking the other results of your paper into account.

Answer: Sorry about the not exact expression and thanks a lot for the suggestions. The other reviewer also suggested that data of Europe and East Asia are not closely related to the HTP. Therefore, we will just use the data of East Asia to prove that BC deposition

of lake sediment of this area can reflect actual atmosphere deposition, but not classify them as "polluted" or "less polluted" areas.

- 340: re much ?

Answer: Sorry about this mistake. It should be "*were much*" and modified in the revised MS.

- 338-344: a usual structure for a paper is not to propose an interpretation at the beginning, but to describe the results and then to draw conclusion! Do you think that you have to take into account some other parameters such as the evaporation of the lake that depends on temperature, surface and depth ?

Answer: Thanks a lot for the suggestion. A sentence "*although the influence of sediment focusing on BC deposition in lake cores has been noted in other areas (Blais and Kalff, 1995; Yang, 2015)*" has been added as the first sentence to change the structure.

- 343: what is NMC09 ?

Answer: NMC09 is a lake core named by the adopted reference (Wang et al., 2011). Therefore, "Fig. 5" was added to the end of the sentence to show its location and reference.

- 347: why the BC concentration in PM2.5 should be lower than in TSP? is this statement is clear for MD due to their large size, it is not directly applicable to BC, soot being usually quite small particles. The following conclusion has therefore to be more seriously validated.

Answer: Thanks for the valuable question. It is natural that BC is emitted from combustion activities and large amount of BC exist in $PM_{2.5}$. However, BC will absorb on the large particle during transportation so that some part of BC will also exist in particles large than $PM_{2.5}$. For instance, BC concentration of $PM_{10}$ of urban cities of Helsinki, Finland (Viidanoja et al., 2002) and Lhasa, China (Li et al., 2016) are higher than those of $PM_{2.5}$. Therefore, this phenomenon must exist in remote area like studied stations. Therefore, a sentence "*Because $PM_{2.5}$ does not include all particles in the atmosphere, the actual BC concentration in the atmosphere should be higher than that of $PM_{2.5}$ (Li et al., 2016; Viidanoja et al., 2002)*" was added into the revised MS.

- 352-355: how can you conclude that the atmosphere and the lake core should have a similar concentration of BC ? This is based on 2 not proved inputs. Moreover 65% «< 100%!

Answer: Sorry about the incorrect expression. "*concentration*" should be "*deposition*" and was modified in the revised MS.

- Figure 5: the axes are not readable.

Answer: The axes of Figure 5 have been modified.

- 407: you probably want to compare BC in ice with BC in atmosphere ? please change the sentence.

Answer: Thanks a lot for the suggestion. This sentence was modified to "*BC deposition rates derived from ice cores and snowpits are proposed to be closer to the actual atmospheric values in the HTP.*"

- 418: the values of BC in remote areas depend on the sources and long range transport processes and cannot be compared without caution.

Answer: Thanks a lot for the suggestion, which the other reviewer also pointed out. Therefore, comparison between BC deposition of the HTP and other remoter areas were deleted from the revised MS. In addition, the last sentence of the paragraph was modified to "*In summary, despite some uncertainty associated with the remote study area, the atmospheric BC deposition rate of 17.9±5.3 mg m-2 a-1 in the glacial region of the HTP is proposed.*" in the revised MS.

- 423: it is perhaps better to give the discrepancy in % instead of in mass concentration

Answer: The value was modified to "*17.9±5.3 mg m$^{-2}$ a$^{-1}$*" in the revised MS.

- 466: if the lake are smaller, the catchment inputs are probably also smaller of the flow through the lake is larger involving a smaller deposition into the lake core. It is therefore not obvious that smaller lake with have a higher BC concentration due to catchment inputs.

Answer: Thanks a lot for the suggestion. To reduce the uncertainty the sentence "The lake cores examined in this study were drilled from the two large Qinghai and Nam Co Lakes, and lakes of this size should only be slightly influenced by catchment inputs. Therefore, the catchment inputs into smaller HTP lakes should be more intense, which should be considered in future studies. " was deleted from the revised MS.

**References:**

[revised manuscript text omitted]

---

## Author Comment (AC2) · 22 Jul 2017

Dear reviewer:

We are grateful for your constructive suggestions and questions, which greatly improve this MS. Despite some critical comments from both reviewer, you give us the overall positive assessments. You asked several important questions on the MS and even suggest the adjustment of MS structure. Meanwhile, we also modified some mistakes that found by ourselves during the modification. In addition, the English of the MS has been improved by the professional English editor of Springer nature (Receipt code: GOTRE-F49-0710225832). According to suggestion of English editor, the title of the MS was modified to "*Reevaluating black carbon in the Himalayas and the Tibetan Plateau: concentrations and deposition*" and all the English of the MS was greatly improved.

We show our great thanks to all the questions and suggestions and have answered all of them. Our answers and modifications in the revised MS were marked in blue. The sentence added in the revised MS was marked in red and italic.

Because new experiment has been conducted by help of other researchers from Shandong University, three more researchers were invited as co-authors of this MS.

If you have more questions or suggestions please let us know.

Best wishes!

Chaoliu Li on behalf of all the co-authors

2017/7/20

**Point-by-point response to reviewer's comments**
**Anonymous Referee #2**

The authors address an important and underappreciated issue: the effect of mineraldust, more precisely inorganic carbonatic carbon, on black and elemental carbon measurements. The authors present some data on this, but the issue is in the current state of the manuscript not discussed thoroughly enough. Secondly, the authors discuss the role of catchment area influx of BC to lake sediment records, causing potentially flawed inferred atmospheric BC deposition results for the studied area. This second hypothesis of the manuscript is poorly     justified and unnecessary for the manuscript.
Answer: Thanks a lot for the very detailed review on the MS and the totally positive attitude to the MS. We divided the comments into different parts for the purpose of answering suggestions and questions more clearly.
- The study discusses two quite separate issues, which should be clarified notably in the introduction: First, it is discussed that (apparently mostly) atmospheric measurements of BC (more specifically EC) may be overestimated for the HTP due to mineral dust interfering the measurements. Secondly, a quite unrelated issue of lake sediment records being affected by riverine influx of BC and not only atmospheric BC deposition.

These both factors may result in over-estimations of BC or EC concentrations in these records. These two study premises should be clearly pointed out. While the hypothesis on atmospheric measurements may be justified, the authors show rash and quite poorly grounded critique on the HTP lake sediments as records of atmospheric BC deposition, as discussed below.

Answer: Thanks a lot for the suggestion. First, the sentence "*Therefore, in this article, we discuss the actual concentrations and deposition of BC in the HTP in order to present basic input data for other important studies on the sources, radiative forcing patterns and chemical transport of BC in this region.*" was shown at last sentence of the first paragraph.

Sorry for not showing very clearly the reason we discuss BC deposition achieved from the lake core sediments in the HTP in this study. The premise of that study of Nam Co lake was declared that the result reflected the atmospheric deposition (The title: "Historical Trends of **Atmospheric Black Carbon** on Tibetan Plateau as Reconstructed from a 150-Year Lake Sediment Record"), which was incorrect and widely adopted by other researches, so that it is need to be pointed out. More importantly, the actual BC deposition of the HTP are still poorly constrained so far and also need to be discussed. Therefore, a sentence "*although the influence of sediment focusing on BC deposition in lake cores has been noted in other areas (Blais and Kalff, 1995; Yang, 2015), it has not been pointed out and evaluated in the HTP*" Was added in the revised MS.

- The language in general is not of the required high quality (particularly in the beginning of the manuscript), and in some places even poor leading to possible misunderstandings.

Answer: The other reviewer has also pointed out the poor English of the MS. The English of the whole MS has been improved by professional English editors. Please check whether the English has reached the level of the ACP this time.

The word "BC deposit" is used incorrectly as a noun (e.g. line 61). It is BC causing the climate impacts, not the environmental record or matrix (i.e. "deposit", as the authors incorrectly use the term) itself.

Answer: Sorry about the carelessness expression. "BC deposit" should be "BC", which has been modified in the revised MS.

- The statements on lines 71-74: "However, the above studies present limitations because of unique environments found in the HTP (e.g., high mineral dust (MD) concentrations in aerosols and catchment inputs to lake sediment). Therefore, the above studies should be re-investigated to better define the actual BC values." are key to the manuscript but very poorly justified by the authors. For instance, the authors cite Kaspari et al., 2011 as being one of the studies that has limitations due to e.g. high mineral dust concentrations in the HTP. However, Kaspari et al. specifically use a BC quantification method (SP2, i.e. single-particle soot photometer) which is specifically NOT influenced by dust. Furthermore, Kaspari et al. (2011) measured mineral dust separately in the same study by using iron as a proxy for mineral dust. Therefore, these statements by the authors are unjustified, and questions also the other citations by the authors. Significantly higher accuracy is required by the authors before making such

bold statements.

Answer: Thanks a lot for your careful review and instructive comments. Study of Kaspari et al. (2011) does not related the influence by dust, which is connect "transportation" that mentioned in the sentence. Sorry for the imprecise reference citation. All of those references not related to aerosol of the HTP were deleted. To make the total expression consistent, the sentence was modified to "*numerous studies have been conducted on the BC concentrations in the atmosphere (Cong et al., 2015; Marinoni et al., 2010; Ming et al., 2010; Wan et al., 2015; Zhao et al., 2013b) and atmospheric BC deposition as determined from lake core sediments (Cong et al., 2013; Han et al., 2015). However, all of these studies exhibit limitations because of certain special environmental factors in the HTP (e.g., high concentrations of mineral dust (MD) in aerosols and catchment inputs to lake core sediment).*" In the revised MS.

- The authors are correct in stating (lines 78-85) that inorganic carbon (IC) may influence the total carbon (TC), particularly organic carbon (OC), and even elemental carbon (EC, thermal optical proxy for BC) concentrations. However, this applies ONLY for OC/EC measurements with the thermal optical transmittance/reflectance (TOT/R) method. Furthermore, it depends on the used protocol for the measurements. Cavalli et al. (2010) have studied this issue thoroughly: "Natural calcite evolves in the He-mode with the EUSAAR_2 and the NIOSH protocol, and will be detected as OC, whereas it evolves in the He/O2-mode with the IMPROVE protocol and will possibly be detected as EC. Neither the NIOSH protocol nor the IMPROVE protocol address definitely this issue of CC. The NIOSH method 5040 recommends fumigation of the aerosol samples with HCl prior to thermal-optical analysis to eliminate any contribution of CC to OC and/or EC signal. However, fumigation with HCl has been shown to cause artificial loss of volatile organic acids (Chow et al., 1993) and to induce intense charring phenomena in ambient aerosol samples (Jankowski et al., 2008). The authors have not addressed these issues appropriately in the beginning of the manuscript. These issues are discussed at the very end of the manuscript which is too late, as readers familiar with these issues may have stopped reading the manuscript after this insufficient introduction.

Answer: Thanks a lot for the suggestions and sorry of not mention those articles mentioned by the reviewer in the MS. Therefore, those three articles (Cavalli et al., 2010; Chow et al., 1993; Jankowski et al., 2008) were added to the introduction part of revised MS. Meanwhile, although fumigation with HCl has been shown to cause artificial loss of volatile organic acids (Chow et al., 1993), the significant relationship between TCO-TCA and Ca (Fig.4) showed this potential loss is weak and IC of mineral dust is the dominating factor. Therefore, a sentence "*Although fumigation with HCl can cause the loss of volatile organic acids in treated samples (Chow et al., 1993), this potential influence is not important because of the significant relationship between $TC_O$-$TC_A$ and Ca (Fig. 4)*" was added into the revised MS.

- One of the main hypotheses of the manuscript is presented on lines 139-155. The authors claim that modelled BC deposition and observed BC deposition in e.g. lake sediments should be of comparable size. As the observed BC deposition in lake

sediments is ca. 10 times higher than the modelled value, the authors claim that the discrepancy should be caused by the lake sediments not representing only BC influx from the atmosphere but also from the catchment area. This is the second main premise of the manuscript, but is unfortunately totally unjustified and false. The statement of the authors that the modelled and observed values should at least be comparable, is false. Previous studies have shown several times models to underestimate observed BC concentration and deposition values (2-5 times or even more than by a magnitude) e.g. in the Arctic and China (e.g. Koch et al., 2009; 2011; Bond et al., 2013 and references therein). These under-predictions by models often relate to difficulties in parameterizing, for instance, BC properties, aging, transportation and scavenging efficiencies properly. Moreover, the authors make a major error: observational data is data that is used to validate modeling results that are based on estimations of BC emission strengths, and not the other way around. Modeling data is validated to be reliable based on observations, and the models are tuned accordingly. Furthermore, the authors make bold presumptions without any scientific evidence of the higher observed BC fluxes in sediment being caused by catchment influx of BC to the sediment cores.

Answer: Thanks for the suggestion. We admit that there are some unreasonable expressions at this part because lots of uncertainties occur in the model results of BC deposition, which cannot be considered as the standard value for comparison. Meanwhile, we will not draw the conclusion in the introduction part that the higher BC deposition derived from lake cores is because of catchment inputs. The reviewer must notice that numbers of other strong evidences were provide at 3.2. part of the MS to prove the catchment inputs of BC to the lake core sediment.

Therefore, large part of that paragraph was cut and modified to the following expression. "*Thus far, only three studies have directly reported on BC deposition in the HTP. One model indicated that the BC deposition in the central HTP was 9 mg m$^{-2}$ a$^{-1}$ (Zhang et al., 2015), which is approximately thirty times lower than the values measured in lake cores at Nam Co and Qinghai lakes (270-390 mg m$^{-2}$ a$^{-1}$) (Fig. 1) (Cong et al., 2013; Han et al., 2011). Although considerable uncertainties exist in atmospheric BC deposition estimated from models (Bond et al., 2013; Koch et al., 2009) and lake core sediments (Cohen, 2003; Yang, 2015), these large differences need to be thoroughly investigated.* ".

As a reviewer, I strongly suggest that the authors should consider getting familiar with basics of paleolimnology before making bold accusations about lake sediments as records of BC deposition, e.g."Paleolimnology" by Andrew S. Cohen (2003, Oxford University Press). Basically, there are lakes that are well suitable as BC repositories as they mainly collect atmospherically deposited material. Crucial is, where the sediment core is collected, at the deepest point of the lake and preferably from a lake with very smooth bathymetry. In such a case the sediment core is not at all affected by redistributed sediment e.g. from the catchment area. So the coring site selection is crucial.

Answer: Thanks a lot for the providing of references, which were added to the revised MS. Yes, as you said, some lake cores are totally not affected by the catchment input. However, as we have proved at 3.2 part, lake core of Nam Co was significantly affected

by catchment input (focusing factor).

Additionally, the lake sediments are dated based on lead210 deposition. As the amount of lead 210 is known for the present, and its halflife time is known, lead210 measurements vertically in a sediment core present a robust dating technique. Subsequently, the amount of lead210 in the sediment samples will also inform the researcher whether the coring site is affected by sediment redistribution or not. If not, then the amount of lead210 is the same in the surface sediment sample as in the atmosphere. If there is less or more, this is an indication of the coring spot losing or gaining additional sediment and thereby also BC (e.g. Blais & Kalff, 1995). This can be studied by the sediment focusing factor of the sediment core (Blais & Kalff, 1995). So have the authors checked what the sediment focusing factor of the studied HTP lakes is? According to their suggestion of the lakes recording 30 times too high BC deposition flux values compared to the modelled values due to sediment influx from the catchment area would mean that the sediment cores would have to receive 30 times more sediment influx from the catchment area than what they accumulate naturally. Such values are unheard of in these types of research. Consequently, the authors need to get a basic understanding of paleolimnology and lake sediments as records of contam-inants before making any such suggestions. Basically, this unfounded premise greatly undermines the half of the manuscript discussing this issue.

Answer: Thanks a lot for the valuable suggestion. However, because the suspended sediment transported by rivers to lake might include both surface soil and old soil. We do not know the exact 210Pb value of that river sediment reach the site of lake core, so that it is hard to use 210Pb of surface lake core sediment to estimate the contribution of river sediment to lake core.

In addition, the following proofs that we have mentioned in the MS have shown strong evidence that lake core sediment was contributed by river sediments.

Firstly, previous study on lake core have revealed that rivers in Nam Co basin transport large volumes of suspended allochthonous sediments into Nam Co Lake annually (Doberschütz et al., 2014). Secondly, previous studies on the accumulation rates of lake cores have revealed significant contributions of riverine particles. For instance, the accumulation rates of Nam Co Lake cores (core NMC 08-1) are consistent with the precipitation variations recorded in the Nam Co Basin over the last 60 years (Fig. 5A) (Wang et al., 2011), which indicates that heavy precipitation promotes the transportation of large riverine particles to the lake, thus increasing the accumulation rates in the lake cores. In addition, the mean grain size of another lake core (core NMC09) showed a significantly positive relationship with precipitation (Fig. 5B), also reflecting the same phenomenon that catchment inputs cause lake core accumulations (Li et al., 2014).

- On lines 156-168 the authors present that other studies made based on ice core records show lower BC concentrations and deposition values than the lake sediment and that ice cores are more suitable than lake sediments to record only atmospherically deposited BC. Yes, ice cores record more reliably only atmospherically deposited BC, but when taking into account the sediment focusing factor, input of BC to the sediment cores from the catchment area can be eliminated and values more reliably presenting

atmospheric deposition can be achieved (Blais & Kalff, 1995). Secondly, the authors make a mistake in comparing directly lake sediment and ice core BC records. This is because BC has been mostly analyzed with different analytical methods from these archives resulting in different types of BC particles being quantified (e.g. Hammes et al., 2007). Subsequently, even from same samples considerably different BC concentrations can be detected (e.g. Watson et al., 2005). Sure, many HTP ice core and lake sediment records measure BC as elemental carbon with thermal-optical methods from the samples so that the values should basically be comparable. However, the thermal-optical measurements of the sediments undergo extensive chemical pretreatment before the actual BC analysis and this may partly result in different BC particles being quantified. The comparison is not as straight-forward as the authors try to convey. Please, also remember to consider the different protocols in the TO-measurements as the authors have pointed out themselves. This issue is, again, discussed at the very end of the manuscript, which is not a proper structure for this.

"Because glaciers are generally located at the highest altitudes of a given region, they only receive wet and dry depositions of BC from the atmosphere." This is a very confusing statement (line 161-162) that needs clarification. Low-elevation glaciers can be found around the world. The key is to collect the ice core from the accumulation area of the glacier, not e.g. ablation zone.

Answer: Thanks a lot for the detailed suggestions. The potential influence of focusing factor to the BC deposition of lake core sediments in the HTP (Blais and Kalff, 1995) was added to of revised MS. For suggestion on the different BC concentrations derived from different methods even for the same materials, detailed explanations were shown in your following question of *lines 323-324, please check it.*

Sorry for the misleading and confused expression of lines 161-162, which was modified to "*Because the cols of glaciers where the snow and ice samples were collected are generally located at the highest altitudes of a given region, BC is only deposited via wet and dry deposition from the atmosphere. Therefore, these data need to be comprehensively evaluated.*"

- I disagree with the comment by the authors (lines 165-168) that "In addition, because the HTP is situated in a remote region, BC deposition patterns in the HTP must be compared to those of other areas (e.g., the Arctic, Europe and eastern China) to better understand the patterns." Europe and especially the Arctic have different sources and deposition processes for BC and comparison of HTP BC deposition with Arctic BC deposition seems unjustified.

Answer: Thanks a lot for the suggestion. The purpose of this sentence in the MS is a little misleading. The original purpose of adopting data of Europe and eastern China is to prove whether the lake core sediments accept BC mainly deposited from the atmosphere. Therefore, the data of East China was kept in the revised MS to prove this idea but not to compare with that of the HTP. Meanwhile, those of deep ocean and Arctic were totally deleted from the revised MS.

- Line 204-205, I hope blank values were subtracted from reported values and not the other way around as suggested in the text.

Answer: Thanks a lot for the suggestion. This expression was modified in the revised MS.

- The tests of the authors to remove carbonates from the atmospheric sample filters by fumigation are valuable and interesting. However, most useful results could have been achieved if these measurements were conducted using all the appropriate different temperature programs available (e.g. NIOSH, IMPROVE and EUSAAR_2), because all these have been used in literature, and carbonatic carbon evolves at different analysis stages during these protocols.

Answer: Thanks a lot for the valuable suggestion. New experiment has been conducted on sixteen acid-fumigated samples to measure BC concentrations with NIOSH and EUSAAR_2 protocols in School of Environmental Science and Engineering, Shandong University to compare with those measured by IMPROVE. BC concentration of transmission signal of these methods were compared. The results showed that firstly, TC concentrations of three methods for the same sample were comparable (Figure R1), which is a normal phenomenon and has been found by previous research (Chow et al., 2001). Secondly, it is obvious that BC (EUSAAR_2) > BC (IMPROVE) > BC (NIOSH) (Figure R2) for studied samples. Ratio of BC(IMPROVE)/BC (NIOSH) of the HTP was $1.36 \pm 0.35$, which loaded within the range of previous reported ratio of 1.2-1.5 (Chow et al., 2001; Reisinger et al., 2008). Meanwhile, ratio of BC(EUSAAR_2) / BC (NIOSH) was $1.88 \pm 0.60$, which was also close to that of 2 found by previous research (Cavalli et al., 2010). Therefore, despite of remote area, the differences of BC concentrations derived from the different methods are similar to that of other regions.

Therefore, A sentence "*To investigate the BC concentration measured by different methods, sixteen acid-fumigated aerosol samples were measured following the EUSAAR_2 and NIOSH protocols for comparison with the results of the IMPROVE protocol. The results showed that the TC concentrations of three methods for the same sample were similar, as suggested by previous research (Chow et al., 2001). The ratios of $BC_{(IMPROVE)}/BC_{(NIOSH)}$ and $BC_{(EUSAAR\_2)}/BC_{(NIOSH)}$ for the studied samples were $1.36 \pm 0.35$ and $1.88 \pm 0.60$, respectively, both of which agreed with the previously proposed ratios of 1.2-1.5 (Chow et al., 2001; Reisinger et al., 2008) and 2 (Cavalli et al., 2010), respectively* " was added to the method part of the revised MS.

[Figure]

Figure R1. Close TC concentrations between IMPROVE and EUSAAR_2 (A), IMPROVE and NIOSH (B).

[Figure]

Figure R2. BC ratio of IMPROVE/NIOSH and EUSAAR_2/ NIOSH.

The discussion on lines 301-310 indicates that the authors aren't really sure what happens to carbonates after the acid fumigation. Does carbonate affect the OC or the EC quantification? Seemingly both, and varyingly from sample to sample. To me, this causes high uncertainties in the interpretations of the data. This procedure: "samples with BCA/BCO above one was set as one in calculation of the average value at two stations" sounds quite artificial and there is a lot of room for intense charring being caused by the fumigation (Jankowski et al., 2008) obstructing the kind of analysis the authors are attempting.

Answer: Thanks a lot for the valuable suggestions and questions.

Yes, the existence of carbonates affects the concentration of both OC and EC because carbonates can be decomposed at both high and low temperature (Karanasiou et al., 2010). The treatment of those aerosol samples "with $BC_A/BC_O$ above one" in the MS is not good. Therefore, those samples with $BC_A/BC_O$ was just shown in Fig.2 but not included in the final calculation. Therefore, the adjusted ratios of $BC_A$ to $BC_O$ for

Nam Co station and Everest station were $0.48 \pm 0.35$ and $0.61 \pm 0.24$, respectively.

Correspondingly, previous reported BC concentrations at two stations were overestimated by approximately 52±35% and 39±24%, respectively. All of these data were added into the revised MS. Finally, the actual BC concentration of these two stations were estimated of 61 ng m$^{-3}$ and 154 ng m$^{-3}$, respectively. It need to point out that BC concentration (127 ng m$^{-3}$) at Nam Co station of a new reference (Zhao et al., 2013a) was used this time due to its longer sample collecting time that that old one.

Meanwhile, we admit that even those with $BC_A/BC_O$ ratio below 1 may influenced by charring of OC during measurement, as suggested by the reviewer (Jankowski et al., 2008).Therefore, a sentence "*Because $BC_A$ cannot be higher than $BC_O$, the samples with $BC_A/BC_O$ values greater than one were not included in the above calculations.*

*Nevertheless, the ratio of BC$_A$/BC$_O$ was considered to be slightly overestimated as some portion of OC was considered BC in the acid-treated samples (Jankowski et al., 2008).*" was added into the revised MS.

Furthermore, to make it more clear. The ratios for TC and BC during monsoon period and non-monsoon period were added on Fig. 2. It is obvious that ratios for both BC and TC during monsoon period were higher than those of non-monsoon period, implying less contributions of mineral dust during monsoon period for both stations due to relatively heavy precipitation.

- The statement on *lines 323-324* "In general, the BC deposition levels measured via different methods should be consistent for a given region." is false for even within environmental matrix, e.g. sediments (Watson et al., 2005; Hammes et al., 2007; Han et al., 2011) or snow samples (e.g. Lim et al., 2014) let alone between different environmental records (e.g. Rose & Ruppel, 2015). It is absolutely inappropriate to compare BC deposition values measured using different analytical methods as these measure different BC particle types, as the previously cited work clearly reveals. By suitable selected work with one another on lines 324-336 the authors were able to compare some similar values recorded with different methods for some regions, but at the same time neglected a huge amount of data available which did not suit this statement.

Answer: Thanks a lot for these valuable suggestions.

We think despite of large uncertainties in comparing BC deposition achieved by different methods, it is still acceptable to do this attempt if high cautions are mentioned.

Although those articles you mentioned discussed the large difference of BC concentrations from different methods, those conclusions were drawn under some conditions that do not fit for this study.

First, it is found "*BC or EC concentrations are found to differ by up to a factor of 7 among different methods; factor of 2 differences are common* (Watson et al., 2005). " Accordingly, BC deposition of Nam Co lake core in our study is over 20 times higher than those of glacial area and model result in the HTP. This ratio is still around 3 times of the above result of (a factor of 7) Watson et al., (2005).

Second, although article Hammes et al., (2007) also found the large differences on BC concentration among different methods (i.e., HCl+HF treatment, $K_2Cr_2O_7/H_2SO_4$), it did not include the method on lake core sediments that we adopted (HCl+HF treatment, IMPROVE-A) (Hammes et al., 2007). BC measurement theories of these two methods were totally different.

Third, while Han et al., (2011) suggested that high-temperature thermal protocols (IMPROVE-A) are suitable for differentiating between soot and other carbon fractions, the purpose of which just prove the advantages of IMPROVE-A method on doing study of lake core sediment (Han et al., 2011).

Fourth, in article of Lim et al., (2014), it was found BC concentration measured from SP2 method was far lower than that of thermal–optical method (Lim et al., 2014) because the former can only measure BC in fine grain size of less than 500 nm (Kaspari et al., 2011). For instance, BC concentrations reported from SP2 (0.6 ng.g$^{-1}$) (Kaspari

et al., 2011) is far lower than that of thermal–optical method (20 ng.g$^{-1}$) for the ice cores of the same glacier (East-Rongbuk glacier) (Ming et al., 2008). To avoid this influence, we did not adopt any BC data from SP2 for comparison in this study.

Although the reviewer pointed out that "it is false to compare BC deposition within the same environmental matrix", but according to our results in this study, the BC deposition data achieved from different periods and derived from three different articles/groups using the same method (thermal-optical analysis) agree very well with each other. In detail, BC deposition of East Rongbuk (10.2 mg m$^{-2}$ a$^{-1}$, the Ströhlein Coulomat 702C and Sunset, the exact method was not shown in the article.) (Ming et al., 2008), Zuoqiupu glacier (12 mg m$^{-2}$ a$^{-1}$) and Muztagh Ata (18 mg m$^{-2}$ a$^{-1}$, DRI, IMPROVE-A) (Xu et al., 2009b) and five glaciers in our previous studies (14.4-25 mg m$^{-2}$ a$^{-1}$, Sunset, NIOSH-5040) (Li et al., 2016a; Li et al., 2016b) are close with each other (Table.2 in the revised MS). Therefore, we think it is fitful of comparing the BC deposition data at glacial region measured by the same method from different studies.

In addition, we also think it is acceptable to compare the BC deposition data of lake core sediments with that of glaciers, despite that the lake core samples experience pretreatment (HCl+HF) before being measured by the same method (IMPROVE-A) with that of glacier samples because the much high recovery (97.6±2.2%) of reference material (marine sediment, NIST SRM-1941b) of pre-treatment method used for lake core sediments (Cong et al., 2013). Similar recovery was also provided by study of Qinghai Lake. For instance, the authors of that study declared that "EC (BC) abundances were only 7% lower as a result of the treatment" for the references tested (3.2 part of (Han et al., 2007)). Therefore, the BC deposition data of lake core sediments and snow samples can be compared despite of large uncertainties. For instance, BC deposition data of Nam Co lake core is over 20 times higher than that of glacier region, which cannot be simply explained by uncertainties of different measurement methods but the catchment input.

Based on the above evidences, the last paragraph of the introduction part was added to show the potential uncertainties of comparison. "*Notably, some uncertainty exists in the comparison of BC data among different studies. Despite recent technological achievements, accurately measuring BC concentrations in ambient samples remains a challenge in atmospheric chemistry research (Andreae and Gelencser, 2006; Bond et al., 2013; Lim et al., 2014). Because the methods used to measure BC concentrations and determine BC deposition levels are not the same, uncertainties will be introduced when directly comparing the results from different studies. For instance, different thermal-optical methods with different temperature increase protocols (e.g., NIOSH vs. IMPROVE vs. EUSAAR_2) will produce different BC concentrations for the same sample (Andreae and Gelencser, 2006; Karanasiou et*

*al., 2015). In general, BC concentrations derived from the IMPROVE method are 1.2-1.5 times higher than those derived from the NIOSH method (Chow et al., 2001; Reisinger et al., 2008), and BC concentrations from the EUSAAR_2 temperature protocol are approximately twice as high as those derived from the NIOSH protocol (Cavalli et al., 2010). Furthermore, lake core samples need be pretreated with HCl and HF several times prior to measurements with the thermal-optical methods (Han et al., 2015). However, because of the complex chemical properties of ambient samples, the "best" thermal-optical protocol has not been identified (Karanasiou et al., 2015), and an exact ratio for BC produced from different methods is difficult to determine. Therefore, although the direct comparison of BC concentrations and deposition levels across different studies presents certain uncertainties in this study, the comparison between lake core and snowpit data is still reliable. For instance, although large uncertainties exist for BC concentrations within the same environmental matrix (Hammes et al., 2007; Han et al., 2011; Watson et al., 2005), the similarity of the BC deposition values among different glaciers (Table 1) in different studies implies that comparing BC deposition data is feasible for the glacial region in the HTP. In addition, because BC concentrations measured via the SP2 method are far lower than those measured via thermal–optical methods (Lim et al., 2014) (the former can only measure BC in grain size finer than 500 nm (Kaspari et al., 2011)), SP2-based BC data were avoided in this study. Furthermore, BC concentrations among different methods have been found to vary by up to a factor of 7 (Watson et al., 2005). Accordingly, the BC deposition in the studied lake core (Nam Co) in this study is estimated to be 20 times higher than those in snowpit and ice core studies in the HTP, providing strong evidence of their differences.*"

- Section "3.2.1. Overestimated BC deposition from lake cores of the HTP" contains a lot more promising discussion that expected on the false presumptions presented in the introduction. This section could be clarified and made more convincing with inclusion of the sediment focusing factor calculation and discussion from the respective lakes. However, it is unclear why the authors want to compare in Table 2 BC deposition results from all around the world to HTP values? It's inappropriate and it seems that the authors have simply selected sites that support their points and are not presenting all available data. For instance, all Greenland ice core data is neglected, although there

is a lot of data available, but it just happens to show almost a magnitude lower BC deposition (e.g. McConnell 2010) than other Arctic results, for which reason it seems not to have been presented here. I strongly suggest leaving away the discussion and data of other areas that HTP and Asia. Particular attention should be paid to the fact with how different methods the results in the different articles are produced. - Also in Section 3.2.2., please remove any comparison of HTP values to e.g. Arctic or European BC deposition values. Such comparisons are inappropriate due to different methodologies and very different sources and BC deposition processes in these areas. Furthermore, these comparisons are not necessary for this manuscript and don't lift its significance in any way.

Answer: We agree with the suggestion and deleted all the BC deposition data of Arctic, Europe and deep ocean, which have little relationship with the topic of this study. Accordingly, data of East Asia were kept in the revised MS to provide supporting evidence for conclusion in the HTP in the MS. Meanwhile, the related data of Arctic, Europe and deep ocean were also deleted in Table.2.

- Surprisingly then, the last paragraph of 3.2.2. discusses the possible uncertainties in comparing the BC deposition results between different methods and environmental archives. Good! Unfortunately, this discussion comes too late in the manuscript and is not thorough enough. Things mentioned earlier in this review should be carefully considered.

Answer: Thanks a lot for the suggestion and this part has been moved to the front part of the revised MS. In addition, more related information of references (Hammes et al., 2007; Lim et al., 2014; Watson et al., 2005) mentioned by reviewer was added to this part to make the discussion and the MS stronger. In detail, this part was modified to the last paragraph of introduction part (have been shown in the above question) and new information from question of *lines 323-324 was also merged into this paragraph.*

All in all, the manuscript contains some important perspectives and a little new data on carbonatic carbon in atmospheric BC samples, but it is questionable whether this is enough data to constitute sufficient scientific novelty for a publication. The amount of carbonatic carbon is measured in some atmospheric samples, but it is not estimated whether or how this would have affected e.g. previous snow and ice core measurements in the HTP, where dust is frequently present. This discussion is majorly hampered by the fact that dust, and therefore carbonate, concentrations vary substantially from sample to the next and will have variable influence accordingly. Much more discussion on these issues is required for the manuscript and even after that, it's maybe enough to publish only as a technical note.

Answer: Thanks a lot for the suggestion on the deeper discussion and the implications of the results of this study. As the reviewer assumed, we also think the contribution of carbonates to BC of snowpit and ice core samples should exist, even larger than that of aerosols at some areas of the THP (e.g., north part of the HTP). Therefore, the following part is added to the discussion part. "*Therefore, the overestimation of BC values is likely greater in the northern and western parts of the HTP than near Nam Co, as we noted previously. MD concentrations have been shown to be much higher than BC*

*concentrations in snow and ice core samples from the HTP (Li et al., 2017; Qu et al., 2014). However, numerous studies have measured BC concentrations without using an acid pretreatment step (Ming et al., 2009; Qu et al., 2014; Wang et al., 2015; Xu et al., 2009a; Xu et al., 2009b). Therefore, the contribution of carbonates in MD to the BC concentrations in snow and ice core samples is likely considerable and needs to be quantitatively evaluated in a future study. Similarly, related HTP studies on other issues, such as BC radiative forcing and atmospheric transport models, based on in situ BC concentrations must be adjusted.*"

The discussion on BC deposition values reported for the HTP in lake sediments not reliably representing atmospheric BC deposition, and that the BC concentrations and deposition should be re-evaluated in the HTP, is unjustified and poorly researched. The fact that two lake sediment records show different deposition values than for instance ice cores and atmospheric observations is ultimately not surprising. If the authors wish to further study this issue they should first try to deeply understand the factors controlling BC deposition in lake sediments and the importance of different analytical methodologies for the interpretations. All discussion on other than HTP, or Asian, BC deposition results should be removed from the manuscript. In my personal opinion, this lake sediment discussion should be completely removed from the manuscript. It is unrelated to the carbonate issue, and there is so much other, e.g. ice core, BC deposition data available from HTP, that it is unnecessary to try to re-evaluate the whole HTP BC deposition patterns based on these two sediment records showing different results than other records in the larger area.

Consequently, the authors present too little own new data and their discussion on literature data is mostly inadequate and not presented in a clear structure in the manuscript.

Answer: We have tried to add data and discussion in the revised MS according to the suggestions of reviewers. We think the scientific significance of the revised MS are enriched and reach the levels of the journal. The innovative idea of this MS is we pointed out two simple but important issues (BC concentration and deposition in the HTP that are overlooked by many previous articles), which will cause rethink of BC research in the HTP for all the related researchers.

References provided by the reviewer:

Blais, J. M; Kalff, J. (1995) The influence of lake morphometry on sediment focussing. Limnol. Oceanogr., 40 (3), 582-588.

Bond, T. C., et al. (2013) Bounding the role of black carbon in the climate system: A scientific assessment. J. Geophys. Res.-Atmos., 188, 5380–5552.

Cavalli, F., et al. (2010) Toward a standardised thermal-optical protocol for measuring atmospheric organic and elemental carbon: the EUSAAR protocol. Atmos. Meas. Tech., 3, 79–89, doi:10.5194/amt-3-79-2010.

Chow, J. C., et al. (1993) The DRI Thermal/Optical Reflectance carbon analysis system: Description, evaluation, and applications in U.S. air quality studies, Atmos. Environ., 27A, 1185–1201.

Hammes, K., et al. (2007) Comparison of quantification methods to measure

firederived
(black/elemental) carbon in soils and sediments using reference materials from soil, water, sediment and the atmosphere. Glob. Biogeochem. Cycles, 21, GB3016, doi: 10.1029/2006GB002914.

Han, Y. M.; et al. (2011) Comparison of elemental carbon in lake sediments measured by three different methods and 150-year pollution history in Eastern China. Environ. Sci. Technol., 45, 5287-5293.

Jankowski, N., et al. (2008) Comparison of methods for the quantification of carbonate carbon in atmospheric PM10 aerosol samples, Atmos. Environ., 42, 8055–8064.

Kaspari, S. D., et al. (2011) Recent increase in black carbon concentrations from a Mt. Everest ice core spanning 1860-2000 AD. Geophys. Res. Lett., 38, L04703, doi:10.1029/2010GL046096.

Koch, D., et al. (2009) Evaluation of black carbon estimations in global aerosol models. Atmos. Chem. Phys., 9, 9001–9026.

Koch, D., et al. (2011) Coupled aerosol-chemistry-climate twentieth century transient model investigation: Trends in short-lived species and climate responses. J. Climate, 24, 2693–2714, doi:10.1175/2011JCLI3582.1.

Lim, S. et al. (2014) Refractory black carbon mass concentrations in snow and ice: method evaluation and inter-comparison with elemental carbon measurement, Atmos. Meas. Tech., 7, 3307–3324. McConnell, J. R. (2010) New Directions: Historical black carbon and other ice core aerosol records in the Arctic for GCM evaluation. Atmos. Environ., 44, 2665–2666.

Rose, N. L., Ruppel, M. (2015) Environmental archives of contaminant particles, Blais JM, Rosen MR, Smol JP (eds.), Environmental Contaminants. Developments in Paleoenvironmental
Research 18, Springer, Dordrecht, pp. 187–221.

Watson, J. G., et al. (2005) Summary of organic and elemental carbon/black carbon analysis methods and intercomparisons. Aerosol Air Qual. Res., 5, 65–102.

[revised manuscript text omitted]

---

## Author Response (AR2)

Dear Editor:

Thanks a lot for your suggestion and positive evaluation on this article. We have answered all the questions of editor and reviewer, please check them. If you have any more suggestion or questions please let us know. Meanwhile, we also made some other minor modifications of expression and updated the acknowledgement part in the revised MS. Our answers and modifications in the revised MS were marked out.

"Seze" has been changed to "size" in Fig.5 of revised article.

Thanks a lot for your work and best wishes.

Chaoliu Li on behalf of all the co-authors

2017/8/30

Suggestions for revision or reasons for rejection (will be published if the paper is accepted for final publication)

- Line 74-77 :It would be good to say what is really discussed, namely the real subjects of the paper.

Answer: Thank for the suggestion. This part was modified to "Therefore, in this article, we discussed the actual concentrations and deposition of BC in the HTP based on data of aerosols collected at two remote stations and previously reported BC deposition data."

- Line 111: why a reference to Fig. 1?

Answer: Fig.1 was deleted and the sentence was modified to "although neither of these two studies have directly discussed this issue or evaluated the effects of IC".

- Lines 130-136: Why the review paper of Ming et al. (An overview of black carbon deposition in High Asia glaciers and its impacts on radiation balance, Jing Ming, Cunde Xiao, Zhencai Duc, Xingguo Yang, Advance in water resources, http://dx.doi.org/10.1016/j.advwatres.2012.05.015) about the BC deposition in Asia Glacier is not cited ?

Answer: Thanks for the suggestion. We double checked this article and found although the title of this article was related to "black carbon deposition", this article only reported the black carbon concentration at glacier region but not black carbon deposition. Because this is an important article on black carbon at glacier region in the HTP, we added this reference at introduction part (lines 116-117) in the revised article.

- Lines 169-170: what is the reason for which this comparison remains reliable ?

Answers: Thanks a lot for the suggestion. This sentence was modified to "the comparison between data of lake core and snowpit is still reliable because BC deposition of the former was much higher (approximately 20 times) than that of the latter, far above up to 7 times among different methods (Watson et al., 2005)."

(Watson et al., 2005)

- Lines258-259: revise the language

Answers: This sentence was modified to "In this study, it was shown that carbonate carbon significantly contributes to the BC, TC and OC concentrations of the TSP samples of Nam Co and Everest stations after comparing BC and OC concentrations between original and acid treated samples.".

- Lines 268-270: This sentence does not say why BC is more influenced than OC and TC. I

45  suppose that BC decompose at higher T than OC and is therefore more influenced.

Answers: Thanks for the suggestion. This sentence was modified to "As proposed in previous work (Chow and Watson, 2002), BC concentrations are more heavily influenced than OC and TC concentrations because carbonates are more prone to decompose at high temperatures along with BC during analyses."

[revised manuscript text omitted]

**1   Introduction**

The Himalayas and the Tibetan Plateau (HTP) region is the highest mountain-plateau system in the world and is the source of approximately ten large rivers in Asia. This region is also sensitive to climate change (Bolch et al., 2012;Kang et al., 2010;You et al., 2010). Black carbon (BC) in and around the HTP have been found to play key roles in climate change patterns in the HTP and Asia, including causing atmospheric warming (Xu et al., 2016;Ramanathan and Carmichael, 2008;Lau et al., 2010;Ji et al., 2015), promoting HTP glacial retreat (Xu et al., 2009;Qu et al., 2014;Li et al., 2017;Zhang et al., 2017b;Ming et al., 2009;Ming et al., 2013), altering monsoon system evolution (Bollasina et al., 2008) and affecting the fresh water supplies of billions of residents across Asia. To date, numerous studies have been conducted on the BC concentrations in the atmosphere (Zhao et al., 2013b;Ming et al., 2010;Cong et al., 2015;Marinoni et al., 2010;Wan et al., 2015) and atmospheric BC deposition as determined from lake core sediments (Han et al., 2015;Cong et al., 2013). However, all of these studies exhibit limitations because of certain special environmental factors in the HTP (e.g., high concentrations of mineral dust (MD) in aerosols and catchment inputs to lake core sediment). Therefore, the above studies should be reinvestigated to better define the actual BC values in the HTP. Therefore, in this article, we discussed the actual concentrations and deposition of BC in the HTP based on data of aerosols collected at two remote stations and previously reported BC deposition data. 
[revised manuscript text omitted]

Ji, Z., Kang, S., Cong, Z., Zhang, Q., and Yao, T.: Simulation of carbonaceous aerosols over the Third Pole and adjacent regions: distribution, transportation, deposition, and climatic effects, Climate Dynamics, 45, 2831-2846, 10.1007/s00382-015-2509-1, 2015.

Jurado, E., Dachs, J., Duarte, C. M., and Simo, R.: Atmospheric deposition of organic and black carbon

700    to the global oceans, Atmospheric Environment, 42, 7931-7939, 2008.

Kang, S., Xu, Y., You, Q., Flügel, W.-A., Pepin, N., and Yao, T.: Review of climate and cryospheric change in the Tibetan Plateau, Environmental Research Letters, 5, 015101, 2010.

Karanasiou, A., Diapouli, E., Cavalli, F., Eleftheriadis, K., Viana, M., Alastuey, A., Querol, X., and Reche, C.: On the quantification of atmospheric carbonate carbon by thermal/optical analysis protocols,

705    Atmospheric Measurement Techniques, 4, 2409-2419, 10.5194/amt-4-2409-2011, 2011.

[revised manuscript text omitted]

Zhao, S., Ming, J., Sun, J., and Xiao, C.: Observation of carbonaceous aerosols during 2006-2009 in

855    Nyainqntanglha Mountains and the implications for glaciers, Environmental Science and Pollution Research, 20, 5827-5838, 10.1007/s11356-013-1548-6, 2013a.

Zhao, Z., Cao, J., Shen, Z., Xu, B., Zhu, C., Chen, L. W. A., Su, X., Liu, S., Han, Y., Wang, G., and Ho, K.: Aerosol particles at a high-altitude site on the Southeast Tibetan Plateau, China: Implications for pollution transport from South Asia, Journal of Geophysical Research-Atmospheres, 118, 11360-11375,

860    10.1002/jgrd.50599, 2013b.

Zhao, Z. Z., Cao, J. J., Shen, Z. X., Huang, R. J., Hu, T. F., Wang, P., Zhang, T., and Liu, S. X.: Chemical composition of PM2.5 at a high-altitude regional background site over Northeast of Tibet Plateau, Atmospheric Pollution Research, 6, 815-823, 10.5094/apr.2015.090, 2015.

Zhu, L., Xie, M., and Wu, Y.: Quantitative analysis of lake area variations and the influence factors from

865    1971 to 2004 in the Nam Co basin of the Tibetan Plateau, Chinese Science Bulletin, 55, 1294-1303, 2010.